# Proteolytic processing induces a conformational switch required for antibacterial toxin delivery

Nicholas L. Bartelli ®[1], Victor J. Passanisi[1], Karolina Michalska ®[2,3,4], Kiho Song[5], Dinh Q. Nhan[6], Hongjun Zhou[1], Bonnie J. Cuthbert ®[7], Lucy M. Stols[2], William H. Eschenfeldt[2], Nicholas G. Wilson ®[1], Jesse S. Basra ®[1], Ricardo Cortes ®[1], Zainab Noorsher[1], Youssef Gabraiel[1], Isaac Poonen-Honig ®[1], Elizabeth C. Seacord[1], Celia W. Goulding ®[7,8], David A. Low ®[5,6], Andrzej Joachimiak ®[2,3,4,9], Frederick W. Dahlquist[1,5,6] & Christopher S. Hayes ®[5,6] ✉

Many Gram-negative bacteria use CdiA effector proteins to inhibit the growth of neighboring competitors. CdiA transfers its toxic CdiA-CT region into the periplasm of target cells, where it is released through proteolytic cleavage. The N-terminal cytoplasm-entry domain of the CdiA-CT then mediates translocation across the inner membrane to deliver the C-terminal toxin domain into the cytosol. Here, we show that proteolysis not only liberates the CdiA-CT for delivery, but is also required to activate the entry domain for membrane translocation. Translocation function depends on precise cleavage after a conserved VENN peptide sequence, and the processed ΔVENN entry domain exhibits distinct biophysical and thermodynamic properties. By contrast, imprecisely processed CdiA-CT fragments do not undergo this transition and fail to translocate to the cytoplasm. These findings suggest that CdiA-CT processing induces a critical structural switch that converts the entry domain into a membrane-translocation competent conformation.

Contact-dependent growth inhibition (CDI) is a form of inter-bacterial competition mediated by CdiB and CdiA two-partner secretion (TPS) proteins. CdiB is an outer-membrane β-barrel protein that exports CdiA using recognition determinants within the N-terminal TPS transport domain of the effector (Fig. 1a, b)[1]. CdiA remains tethered to the cell surface, where it is presented to bind neighboring bacteria and deliver its toxic C-terminal region (CdiA-CT) to inhibit target-cell growth (Fig. 1b)[2]. Auto-inhibition is prevented through the expression of cognate CdiI immunity proteins, which specifically neutralize CdiA-

CT toxicity. Many important bacterial pathogens harbor *cdi* gene clusters, and CDI activity has been demonstrated for *Escherichia coli*[3], *Dickeya dadantii*[4], *Neisseria meningitidis*[5], *Enterobacter cloacae*[6], *Pseudomonas aeruginosa*[7–9], *Acinetobacter baumannii*[10,11], and several *Burkholderia* species[12–15]. CDI systems are notable for the variety of toxins they deploy, with at least 130 distinct CdiA-CT sequence types recognized[4,16]. In the Enterobacteriaceae, the polymorphic CdiA-CT region is usually delineated by a conserved VENN peptide motif from the adjacent pretoxin-VENN (PT-VENN) domain (Fig. 1a)[4,17]. CdiA-CTs

[1]Department of Chemistry and Biochemistry, University of California, Santa Barbara, CA, USA. [2]Midwest Center for Structural Genomics, Argonne National Laboratory, Lemont, IL, USA. [3]Center for Structural Genomics of Infectious Diseases, University of Chicago, Chicago, IL, USA. [4]Structural Biology Center, X-ray Science Division, Argonne National Laboratory, Lemont, IL, USA. [5]Biomolecular Science and Engineering Program, University of California, Santa Barbara, CA, USA. [6]Department of Molecular, Cellular and Developmental Biology, University of California, Santa Barbara, CA, USA. [7]Department of Molecular Biology & Biochemistry, University of California, Irvine, CA, USA. [8]Pharmaceutical Sciences, University of California, Irvine, CA, USA. [9]Department of Biochemistry and Molecular Biology, University of Chicago, Chicago, IL, USA. ✉e-mail: chayes@lifesci.ucsb.edu

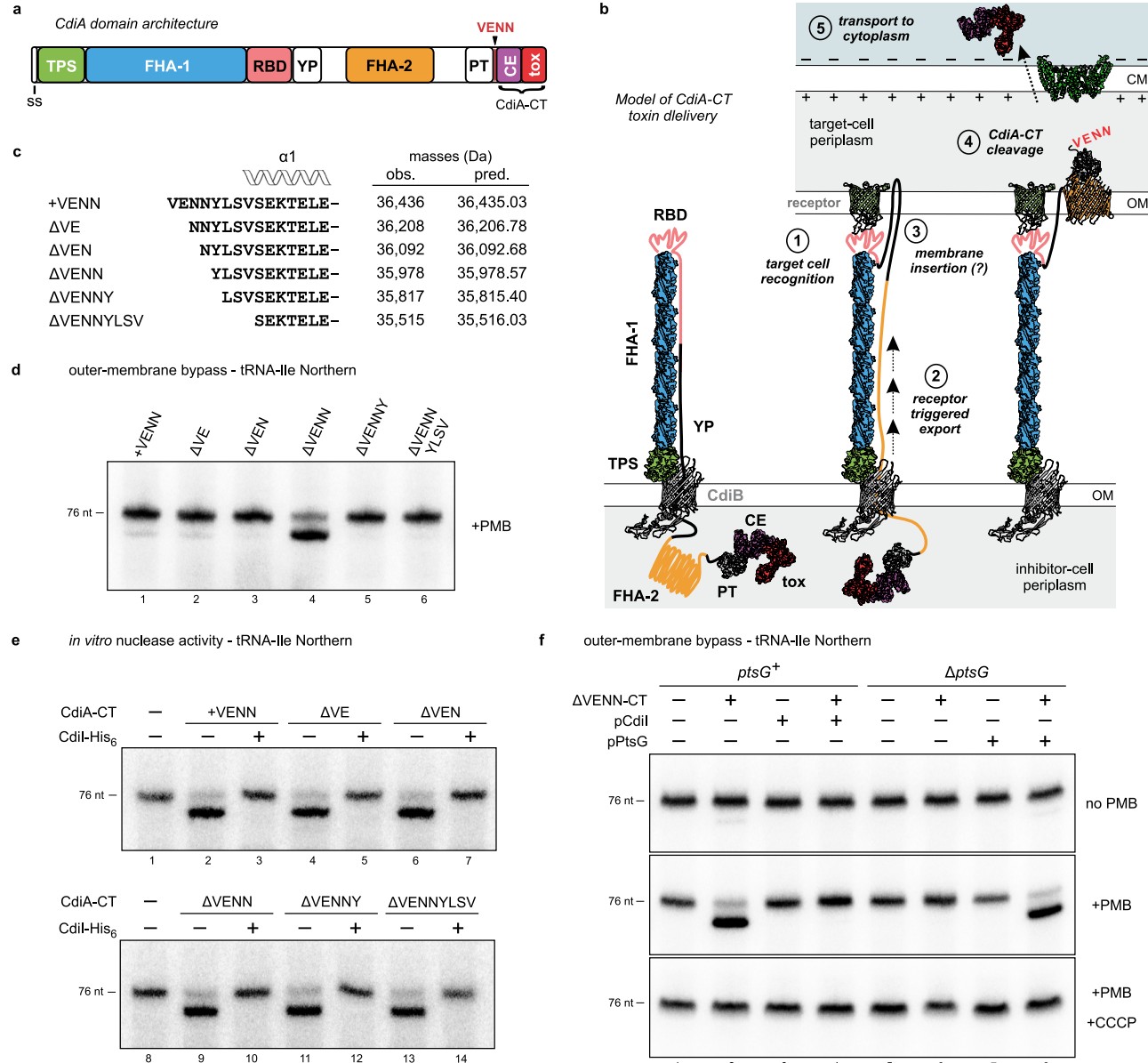

**Fig. 1 | CdiA-CT processing is required for translocation into the target-cell cytoplasm. a** E. coli CdiA protein domain arrangement. **b** Model of CdiA secretion arrest and receptor-triggered toxin delivery. **c** N-terminal sequences of CdiA-CT[EC3006] variants used in outer-membrane bypass and in vitro tRNase assays. **d** Outer-membrane bypass assays. Purified CdiA-CT[EC3006] variants were incubated with polymyxin B (PMB) treated E. coli cells. Total RNA was isolated for Northern blot analysis using a probe to tRNA[Ile]. **e** In vitro nuclease assays. E. coli total RNA was treated with purified CdiA-CT[EC3006] in the absence and presence of CdiI[EC3006]. Reactions were analyzed by Northern blotting using a probe to tRNA[Ile]. **f** CdiA-CT[EC3006] entry depends on PtsG and the proton gradient. Purified ΔVENN CdiA-

CT[EC3006] was incubated with E. coli ΔptsG and ptsG+ cells treated with PMB and carbonyl cyanide-m-chlorophenylhydrazone (CCCP). Where indicated, cells also carried plasmids that encode PtsG and CdiI[EC3006]. Total RNA was isolated and analyzed by Northern blotting using a probe to tRNA[Ile]. Experiments depicted in panels **d**–**f** were repeated independently twice with similar results. CCCP carbonyl cyanide m-chlorophenylhydrazone, CE cytoplasm entry, CM cytoplasmic membrane, FHA filamentous hemagglutinin, nt nucleotides, PMB polymyxin B, PT pretoxin, RBD receptor-binding domain, ss signal sequence, tox toxin, YP tyrosine-proline enriched. Source data are provided as a Source Data file.

are modular and can often be delivered by heterologous CdiA proteins when fused at the common VENN motif[4,18]. This architecture enables the recombination of horizontally acquired cdiA-CT/cdiI sequences to diversify toxin repertoires[5,19]. The extraordinary diversity of CDI toxin-immunity protein pairs, coupled with their frequent horizontal transfer, suggests that the evolution of novel CdiA effectors provides a fitness advantage to bacteria.

Recent studies provide a mechanistic framework for CdiA export and toxin delivery. Electron cryotomography has revealed that each individual CdiA protein forms a thin filament projecting several hundred angstroms from the cell surface[2]. This extracellular structure

corresponds primarily to the filamentous hemagglutinin-1 (FHA-1; Pfam PF05594) repeat domain (Fig. 1a, b), which is predicted to fold into an elongated β-helix[20]. The receptor-binding domain (RBD) of CdiA is located at the distal tip of the filament[2,21]. During biogenesis, CdiA export across the outer membrane is halted when CdiB encounters a Tyr- and Pro-enriched segment adjacent to the RBD (Fig. 1b). Consequently, the entire C-terminal half of CdiA – including the toxic CdiA-CT and another repeat domain composed of distinct FHA-2 (PF13332) motifs – is retained within the periplasm (Fig. 1a, b). CdiA remains in this partially exported state until it engages its receptor, which triggers secretion to resume through an unknown

mechanism. The FHA-2 repeat domain is then secreted onto the target bacterium, where it is thought to form a conduit to translocate toxin across the target-cell outer membrane (Fig. 1b, step 3). After transfer into the target-cell periplasm, the CdiA-CT is released through proteolytic processing (Fig. 1b, step 4), which is likely required for toxin delivery into the target-cell cytosol, because mutation of the VENN motif to VENA prevents CdiA-CT cleavage and abrogates growth inhibition activity[2].

CdiA-CT regions are composed of two domains that have distinct functions during CDI. The extreme C-terminal domain is typically a nuclease responsible for growth inhibition activity, and the N-terminal cytoplasm-entry domain mediates translocation into the target-cell cytosol[18]. The latter transport step also depends on specific cytoplasmic membrane proteins in the target cell (Fig. 1b, step 5). In the Enterobacteriaceae, entry domains that exploit MetI, GltJK, RbsC, FtsH, PtsG, SecY, and YciB have been described[18,22]. A similar import strategy is utilized in other phylogenetic clades. GltJK and a predicted major facilitator superfamily transporter protein are required for the intoxication of *Burkholderia* target cells[23,24], and an ABC transporter of dipeptides (DppBC) is hijacked for CDI toxin import in *P. aeruginosa*[9]. Many entry domains can guide different toxic cargos into target bacteria. For example, CdiA proteins from *E. coli* strains 3006, NC101 and STEC_O31 all contain PtsG-dependent entry domains, though these effectors deliver tRNase toxins with distinct structures and substrate specificities[18,25–27]. Cell import also depends on the proton gradient across the cytoplasmic membrane of target bacteria[28], suggesting that the polarized membrane provides energy to power translocation.

Here, we report that the PtsG-dependent entry domain of CdiA[EC3006] from *E. coli* 3006 undergoes a conformational change in response to proteolytic processing. Crystallography of CdiA-CT[EC3006] carrying an N-terminal VENN sequence shows that the entry domain is composed of nine α-helices organized into two subdomains. However, this +VENN form of CdiA-CT[EC3006] is unable to translocate across the cytoplasmic membrane. Using an outer-membrane bypass approach to introduce purified CdiA-CT[EC3006] variants into the periplasm, we show that the VENN motif must be removed (ΔVENN) to enable transport into the cytoplasm. Moreover, translocation activity is remarkably sensitive to the identity of the newly formed N-terminus. CdiA-CT[EC3006] fragments fail to enter the cytoplasm if they contain even one residue too many (ΔVEN) or too few (ΔVENNY) at the N-terminus. The functional ΔVENN entry domain also exhibits biophysical features that are distinct from improperly processed forms. Together, these findings indicate that precise processing is required to convert the entry domain into a translocation competent state.

## Results

### Structure of the PtsG-dependent cytoplasm entry domain
The structure of CdiA-CT[EC3006] bound to its cognate CdiI[EC3006] immunity protein was solved to 2.25 Å resolution (PDB: 6VEK) (Table 1 and Supplementary Fig. 1). The asymmetric unit contains a single CdiA-CT•CdiI[EC3006] complex, and the refined model includes CdiA-CT[EC3006] residues Glu2-Gly77 and Tyr92-Phe336 (numbered from Val1 of the VENN motif) and residues Ser6-Pro158 of CdiI[EC3006]. CdiA-CT[EC3006] is bipartite and consists of an N-terminal cytoplasm-entry domain and a C-terminal BECR-fold tRNase domain (Fig. 2a). The tRNase domain structure and its interactions with CdiI[EC3006] are similar to those reported previously[25], with rmsd values of 0.94/0.91 Å for the nuclease domain and 0.43/0.44 Å for the immunity protein. We note that the tRNase domain in the previously reported structure (PDB:6CP8) lacks residue Asn337, altering the orientation of active-site residue Tyr208 relative to the wild-type structure presented here (Supplementary Fig. 2a).

The CdiA-CT[EC3006] entry domain consists of nine α-helices organized into two subdomains (Fig. 2a). A DALI server search failed to identify close structural homologs for the entry domain, though

**Table 1 | Crystallography data processing and refinement statistics**

| Data processing | |
|---|---|
| Protein | CdiA-CT•CdiI[EC3006] |
| Wavelength (Å) | 0.9793 |
| Resolution range (Å)[a] | 30.00 – 2.25 (2.29 – 2.25) |
| Space group | $P2_12_12_1$ |
| Unit cell parameters (Å) | 41.00, 71.77, 175.54 |
| Unique reflections | 25,933 (1274) |
| Multiplicity | 25.4 (17.2) |
| Completeness (%) | 99.9 (99.7) |
| $\langle I \rangle / \langle \sigma I \rangle$ | 30.29 (1.77) |
| $R_{merge}$[b] | 0.132 (1.242) |
| $CC_{1/2}$[c] | 0.635 |
| $CC*$[c] | 0.881 |
| **Refinement** | |
| Resolution (Å) | 29.26 – 2.25 |
| Reflections work/test set | 23,763/1652 |
| $R_{work}$/ $R_{free}$[d] | 0.182/0.234 |
| Average B factor (Å$^2$) (No of atoms) | |
| macromolecule | 65.3 (3677) |
| solvent | 51.2 (109) |
| Rmsd bond lengths (Å) | 0.007 |
| Rmsd bond angles (°) | 0.786 |
| Ramachandran favored[e] (%) | 96.79 |
| Ramachandran outliers | 0.21 |
| Clashscore[e] | 2.6 |
| PDB ID | 6VEK |

[a]Values in parentheses correspond to the highest resolution shell.
[b]$R_{merge} = \Sigma_h \Sigma_j \mid I_{hj} - \langle I_h \rangle \mid / \Sigma_h \Sigma_j I_{hj}$, where $I_{hj}$ is the intensity of observation $j$ of reflection $h$.
[c]As defined by ref. 68.
[d]$R = \Sigma_h \mid F_o \mid - \mid F_c \mid / \Sigma_h \mid F_o \mid$ for all reflections, where $F_o$ and $F_c$ are observed and calculated structure factors, respectively. R$_{free}$ is calculated analogously for the test reflections, randomly selected and excluded from the refinement.
[e]As defined by Molprobity[69] and implemented in Phenix.

several proteins contain similar individual α-hairpin motifs. The N-terminal subdomain is composed of α1-α2 and α3-α4 hairpins, with helices α3 and α4 linked through a disulfide bond between Cys55 and Cys65. The segment (Gly77–Tyr92) connecting the subdomains is not resolved in the structure. Helices α5 through α9 comprise the C-terminal subdomain, with α6-α7 and α8-α9 forming α-hairpins. The α8-α9 hairpin of the entry domain interacts with the globular α/β-core of the tRNase domain through several specific contacts. Lys170 in helix α9 forms an H bond with the backbone carbonyl of Val179 and a salt-bridge with Glu178 from the nuclease domain (Fig. 2b). Arg172 forms a buried salt-bridge with Glu206, and another ion-pair links Asp143 and Lys212. tRNase residue Ser281 forms H bonds to Ser174 and Asn175 within the loop connecting the two domains. Finally, the side chains of Tyr142 and Leu173 pack onto hydrophobic patches on the nuclease domain.

The electron density allowed us to model side chains for most of the VENN motif, which lies at the convergence of helices α1, α2, α3, and α4 (Fig. 2c). Asp44 within helix α2 coordinates this nexus through interactions with Asn3, the backbone amide of Tyr5, and the side chain of Lys76 in helix α4 (Fig. 2c). The side chain of Tyr5 also packs against Asp48 at the junction of α2 and α3 (Fig. 2c). We note that the conformation of the VENN peptide may be influenced by crystal packing forces, because each α1-α2 hairpin inserts into the cleft separating the entry subdomains of the neighboring protomer (Fig. 2c and Supplementary Fig. 2b).

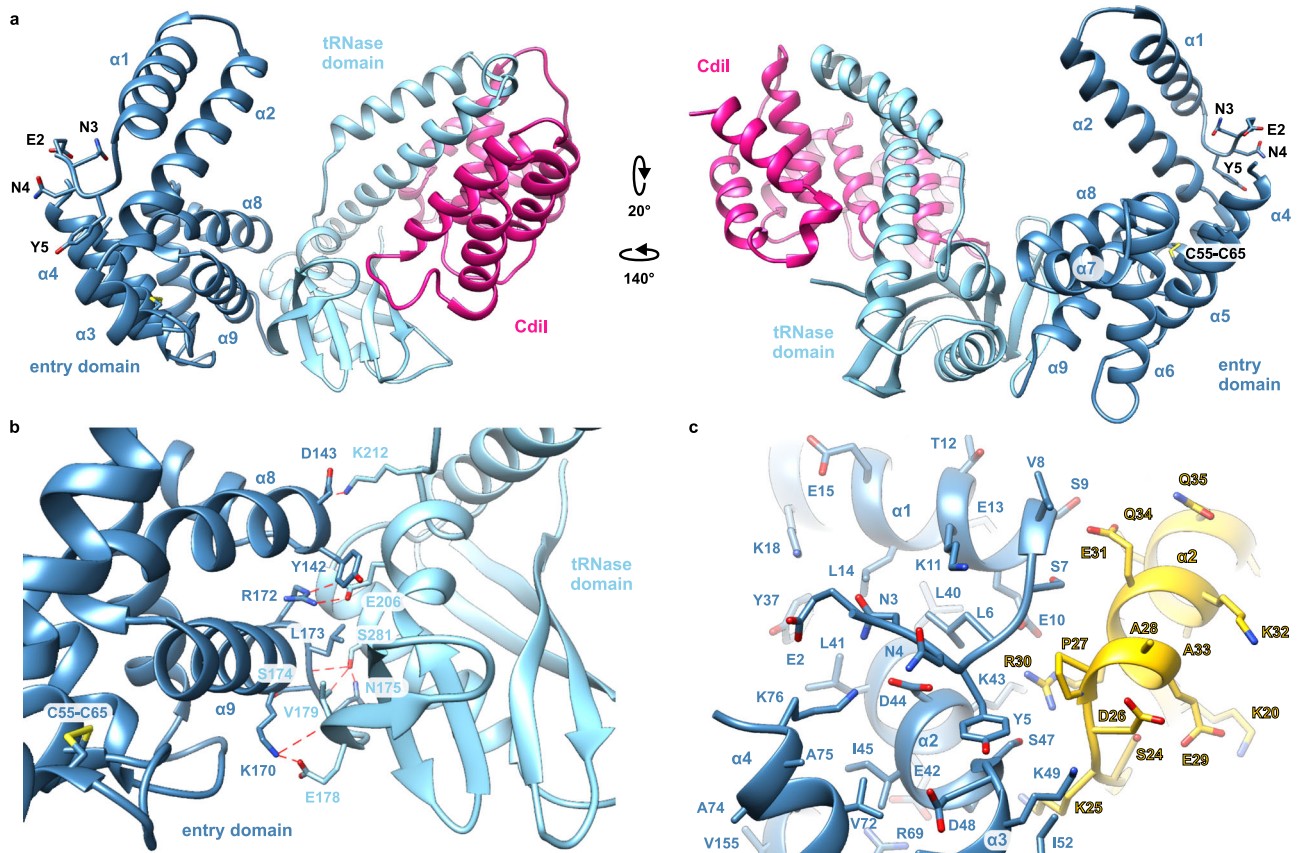

**Fig. 2 | Structure of the PtsG-dependent cytoplasm-entry domain. a** Structure of the CdiA-CT•CdiI$^{EC3006}$ complex. CdiA-CT$^{EC3006}$ is composed of an N-terminal cytoplasm-entry domain (dark blue) and a C-terminal BECR tRNase (light blue). Side chains for Glu2, Asn3, and Asn4 (of the CdiA-CT$^{EC3006}$ VENN motif) Tyr5 and the Cys55·Cys65 disulfide are shown as sticks. **b** Intra-molecular contacts between entry and tRNase domains. **c** Interactions between adjacent entry domains (in dark blue and yellow) in the crystal lattice.

## CdiA-CT processing is required for translocation into the target-cell cytoplasm

CdiA-CT processing is required for growth inhibition activity, suggesting that it must be released into the periplasm before translocation into the cytoplasm. This model predicts that CdiA-CT fragments should be capable of translocation, provided they have access to their membrane receptors. To test this hypothesis, we permeabilized the outer membrane of *E. coli* cells with polymyxin B (PMB), which allows purified CdiA-CT$^{EC3006}$ fragments into the periplasmic space. Translocation to the cytosol was monitored by Northern blot analysis to detect cleaved tRNA$^{Ile}$ from the toxin nuclease activity. We first tested CdiA-CT$^{EC3006}$ carrying the N-terminal VENN motif and observed only trace nuclease activity (Fig. 1d, lane 1). Given the importance of proteolytic processing, we reasoned that the identity of the newly formed N-terminus could be critical for cell entry. Therefore, we generated and tested a series of CdiA-CT$^{EC3006}$ constructs with defined N-termini (Fig. 1c and Supplementary Fig. 3). All CdiA-CT$^{EC3006}$ variants exhibit the same tRNase activity in vitro (Fig. 1e), but only the ΔVENN construct enters the cytoplasm of PMB treated cells efficiently (Fig. 1d, lane 4). This transport corresponds to the CDI delivery pathway, because tRNase activity is only observed in cells that express *ptsG* (Fig. 1f, middle panel, lanes 2 and 8). Furthermore, ΔVENN CdiA-CT$^{EC3006}$ does not enter permeabilized *ptsG*⁺ cells in the presence of carbonyl cyanide-*m*-chlorophenylhydrazone (CCCP), which dissipates the proton gradient (Fig. 1f, bottom panel, lanes 2 and 8). These results show that CdiA-CT$^{EC3006}$ follows its physiological cell-entry pathway when the outer-membrane translocation step is bypassed. Moreover, because only the ΔVENN construct enters the cytosol efficiently, these results

strongly suggest that the CdiA-CT region is normally processed precisely after the VENN sequence.

## CdiA-CTs follow independent, parallel entry pathways

To test whether other cytoplasm-entry domains mediate translocation in the outer-membrane bypass assay, we examined CdiA-CT$^{Ym43969}$ from *Yersinia mollaretii* ATCC 43969, which contains an EndoU RNase domain that cleaves the anticodon loop of tRNA$^{Glu}$ [26]. Because the CdiA-CT$^{Ym43969}$ entry domain is uncharacterized, we used a genetic approach to identify its receptor. *E. coli* cells were subjected to *mariner* transposon mutagenesis, and CDI-resistant mutants were selected in co-cultures with inhibitor cells that deliver CdiA-CT$^{Ym43969}$. All resistant mutants carried transposon insertions in *acrB*, which encodes a membrane-embedded multidrug efflux pump[29]. We confirmed that Δ*acrB* deletion mutants are as resistant to CdiA-CT$^{Ym43969}$ intoxication as *acrB*⁺ target cells that produce CdiI$^{Ym43969}$ immunity protein (Fig. 3a). Δ*acrB* cells also become CDI-sensitive when complemented with the wild-type *acrB* gene (Fig. 3a). Although Δ*acrB* mutants are resistant to CdiA-CT$^{Ym43969}$ delivered through the CDI pathway, their growth is still inhibited when the toxin is expressed in the cytoplasm under control of an arabinose-inducible promoter (Fig. 3b), and internally produced toxin cleaves tRNA$^{Glu}$ in both *acrB*⁺ and Δ*acrB* backgrounds (Fig. 3c, lanes 3 and 6). Together, these findings suggest that AcrB is required for CdiA-CT$^{Ym43969}$ entry into the target-cell cytoplasm. Accordingly, purified CdiA-CT$^{Ym43969}$ enters the cytosol of PMB permeabilized *acrB*⁺ cells, but not Δ*acrB* mutants (Fig. 3d, lanes 6, 7, and 8). Furthermore, the PtsG and AcrB entry pathways are parallel and independent, because purified CdiA-CT$^{EC3006}$ enters permeabilized Δ*acrB* cells (Fig. 3d, lane 3), and CdiA-CT$^{Ym43969}$ enters Δ*ptsG* cells (Fig. 3d, lane 7).

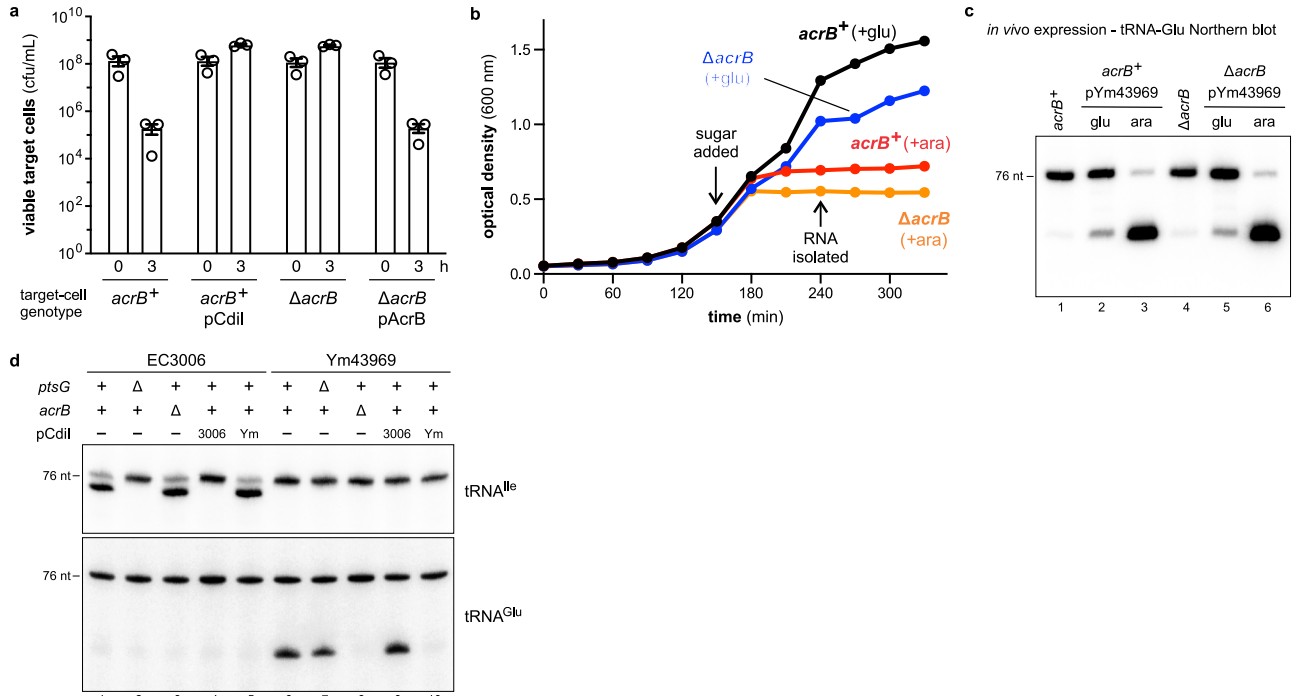

**Fig. 3 | CdiA-CTs follow independent, parallel entry pathways. a** Competition co-cultures. Target bacteria were mixed at a 1:1 ratio with inhibitor cells that deploy CdiA-CT[Ym43969] and viable target cells were enumerated as colony-forming units (cfu) per mL at 0 and 3 h. Presented data were the average ± SEM for three independent experiments. **b** Internal toxin production. *cdiA-CT[Ym43969]* expression was induced with arabinose (+ara) or repressed with glucose (+glu) in *E. coli acrB[+]* and *ΔacrB* cells. Cell growth was monitored by optical density. Presented data are the average for two independent experiments. **c** RNAs isolated from cells in panel **b** were analyzed by Northern blotting using a probe to tRNA[Ile]. **d** Outer-membrane bypass assays. Polymyxin B (PMB) treated *E. coli* cells were incubated with purified ΔVENN CdiA-CT[EC3006] or CdiA-CT[Ym43969], and total RNA was isolated for Northern blot analyses using probes to tRNA[Ile] and tRNA[Glu]. Where indicated, cells also carried plasmids that express *cdiI[EC3006]* or *cdiI[Ym43969]* (pCdiI). The experiment in panel **a** was repeated independently three times, and the experiments in panels **b**–**d** were repeated independently twice with similar results. Source data are provided as a Source Data file.

## The N-terminus of the PtsG-dependent entry domain controls a structural switch

CdiA-CT[EC3006] translocation depends on its precise N-terminal sequence, suggesting that the structure of the ΔVENN entry domain differs from the +VENN form determined by crystallography. Circular dichroism (CD) spectroscopy of the ΔVEN, ΔVENN, and ΔVENNY PtsG-dependent entry domains shows that all variants have similar α-helical content (Fig. 4a). However, nuclear magnetic resonance (NMR) spectroscopy revealed significant differences in the conformation of the ΔVENN domain. $^{1}$H-$^{15}$N heteronuclear single quantum coherence (HSQC) spectra of the ΔVEN and ΔVENNY entry domains are typical of folded proteins with limited conformational dynamics (Fig. 4b). By contrast, the translocation competent ΔVENN entry domain has significantly fewer resonances (Fig. 4b), suggesting dynamic structural interconversion in the millisecond timescale[30]. These spectroscopic features are consistent with a molten globule, which has a native-like secondary structure, but a poorly defined tertiary structure. Because molten globules have disorganized hydrophobic cores, they typically do not produce CD signals at near UV wavelengths where aromatic residues absorb[31,32]. However, the ΔVENN domain exhibits more negative ellipticity at 260−290 nm than the ΔVEN and ΔVENNY variants (Supplementary Fig. 4a). Molten globules are also commonly probed using 8-anilino-1-naphthalenesulfonate (ANS), which undergoes a shift in fluorescence emission upon binding solvent-accessible hydrophobic residues[32]. ANS fluorescence is somewhat enhanced by the ΔVENN domain compared to a well-folded control protein (hen lysozyme), but the emission is very similar to that observed with the ΔVEN and ΔVENNY variants (Supplementary Fig. 4b). Together, these results are not consistent with a molten globule state for the ΔVENN domain.

The lack of $^{1}$H-$^{15}$N HSQC resonances could also reflect protein aggregation. Size-exclusion chromatography shows that the ΔVENN domain does not form stable aggregates, though it elutes from the column earlier than the ΔVEN variant (Supplementary Fig. 5a). Multi-angle light scattering (MALS) analysis of the eluates indicates that each domain is primarily monomeric (Supplementary Fig. 5b, c), but because samples are diluted during chromatography, it is possible that low-affinity multimers dissociate prior to elution. Indeed, the elution profile shifts earlier when more ΔVENN domain is loaded, though MALS indicates little change in mass and hydrodynamic radius (Supplementary Fig. 5d). Given that MALS analysis shows that the ΔVENN domain is primarily monomeric at concentrations up to 20 μM, we collected another $^{1}$H-$^{15}$N HSQC dataset at this lower concentration. The resulting spectrum contains more resonances with better dispersion (Supplementary Fig. 6a), though inspection of the original ΔVENN spectrum at very low contour also reveals these resonances at reduced intensity (Supplementary Fig. 6b). Low-contour overlays of the original high-concentration spectra show a handful of resonances are unique to the ΔVENN form (Supplementary Fig. 6c, d). Together, these results suggest that NMR signal broadening is due to ΔVENN domain oligomerization at high concentrations. These quaternary interactions are low-affinity and dynamic, enabling the domain to revert to a monomeric form upon dilution.

We next examined the thermodynamic stabilities of the entry domain variants. CD spectroscopy revealed cooperative unfolding transitions for ΔVEN and ΔVENNY domains when denatured with urea (Fig. 4c). These two-state processes correspond to unfolding energies ($\Delta G_u$) of 2.2 (±0.1) kcal/mol and 2.2 (±0.2) kcal/mol for the ΔVEN and ΔVENNY entry domains, respectively. By contrast, the ΔVENN entry domain is more resistant to chemical denaturation and undergoes a

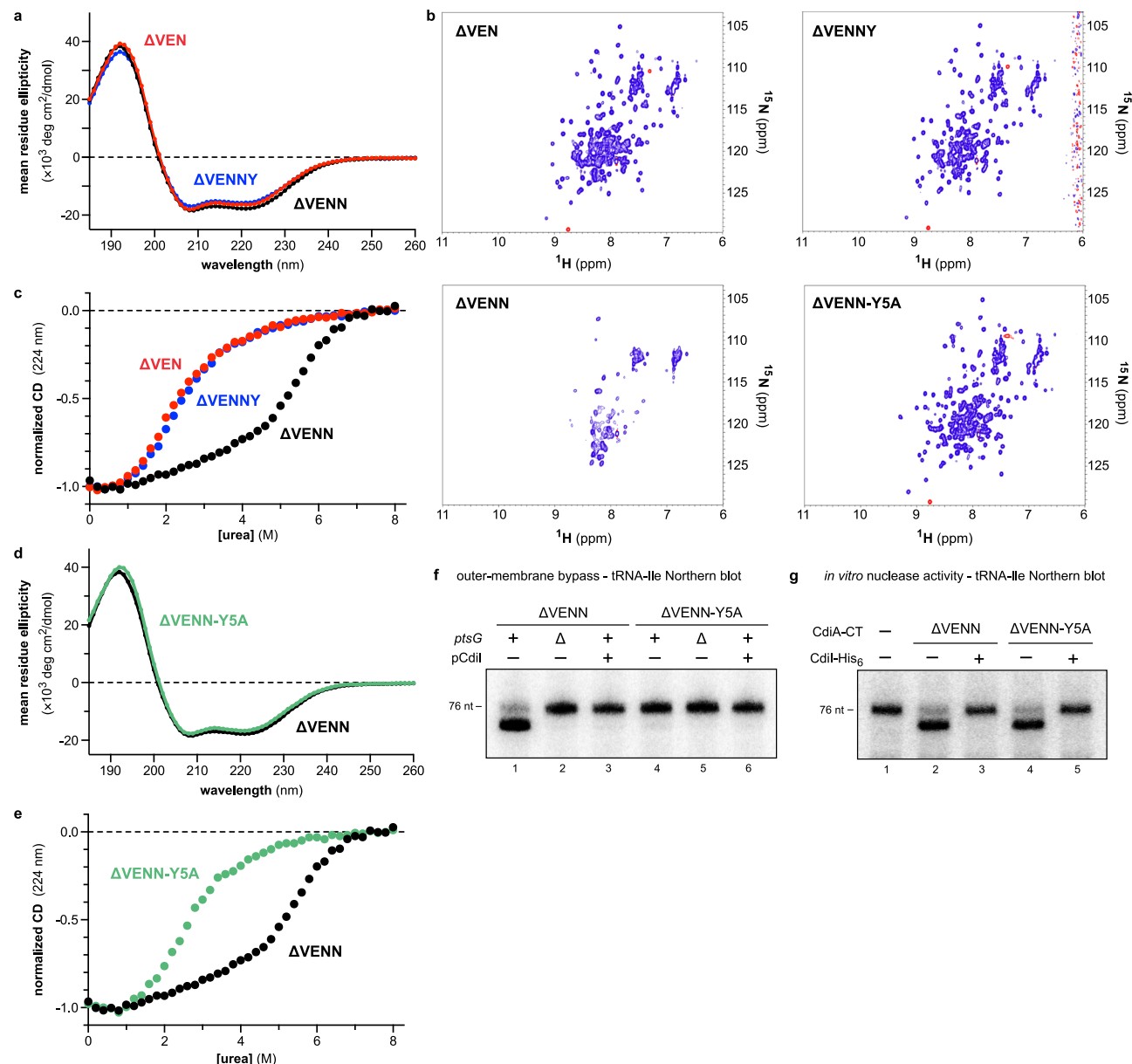

**Fig. 4 | The N-terminus of the PtsG-dependent entry domain controls a conformational switch. a** CD spectra of PtsG-dependent entry domains. **b** $^1$H-$^{15}$N HSQC NMR spectra of PtsG-dependent entry domains. **c** Chemical denaturation of PtsG-dependent entry domains. Purified proteins were denatured in urea and unfolding monitored by CD at 224 nm. **d** CD spectra of ΔVENN and ΔVENN-Tyr5Ala entry domains. **e** Chemical denaturation of ΔVENN and ΔVENN-Tyr5Ala entry domains. **f** Outer-membrane bypass. Purified CdiA-CT$^{EC3006}$ variants were incubated with polymyxin B (PMB) treated *E. coli* cells, and total RNA was isolated for Northern blot analysis using a probe to tRNA$^{Ile}$. Where indicated, cells carried the Δ*ptsG* allele and/or a plasmid that expresses *cdiI*$^{EC3006}$. **g** In vitro nuclease activity. *E. coli* total RNA was treated with CdiA-CT$^{EC3006}$ proteins in the absence and presence of purified CdiI$^{EC3006}$ immunity protein (CdiI-His$_6$). The experiments in panels **a**, **c**, **e**, and **g** were repeated independently twice with similar results. The experiments in panels **b**, **d**, and **f** were performed once. Source data are provided as a Source Data file.

gradual linear transition over lower urea concentrations, followed by a more rapid loss of CD signal at ~6 M urea (Fig. 4c). A two-state unfolding model cannot be fitted to these data, suggesting that denaturation proceeds through multiple conformational states. Importantly, oligomerization cannot account for increased stability because these experiments were performed at sub-micromolar concentrations where the ΔVENN domain is monomeric.

In principle, entry domain conformation could be influenced by the tRNase domain in the context of the intact CdiA-CT$^{EC3006}$. However, $^1$H-$^{15}$N HSQC spectra of CdiA-CT$^{EC3006}$ constructs show the same general features as the corresponding entry domains, with the ΔVENN variant characterized by broadened, overlapping resonances (Supplementary Fig. 7a, b). These NMR features suggest that ΔVENN CdiA-CT$^{EC3006}$ self-

associates at high concentration, though oligomeric forms are not detected by SEC-MALS analysis (Supplementary Fig. 5a, e). We also examined the isolated tRNase domain and found that it has good resonance dispersion (Supplementary Fig. 7c), suggesting that it adopts a well-ordered structure independent of the N-terminal entry domain. A number of tRNase residues, including Trp318 in the hydrophobic core, exhibit equivalent chemical shifts in the CdiA-CT$^{EC3006}$ constructs (Supplementary Fig. 7a, c, d), indicating that the conformational state of the entry domain has little effect on the nuclease fold. Moreover, the tRNase domain does not appear to affect the entry domain during chemical denaturation. Unfolding profiles for the ΔVEN entry domain and CdiA-CT$^{EC3006}$ are very similar (Supplementary Fig. 7e), suggesting that the tRNase and ΔVEN entry domains

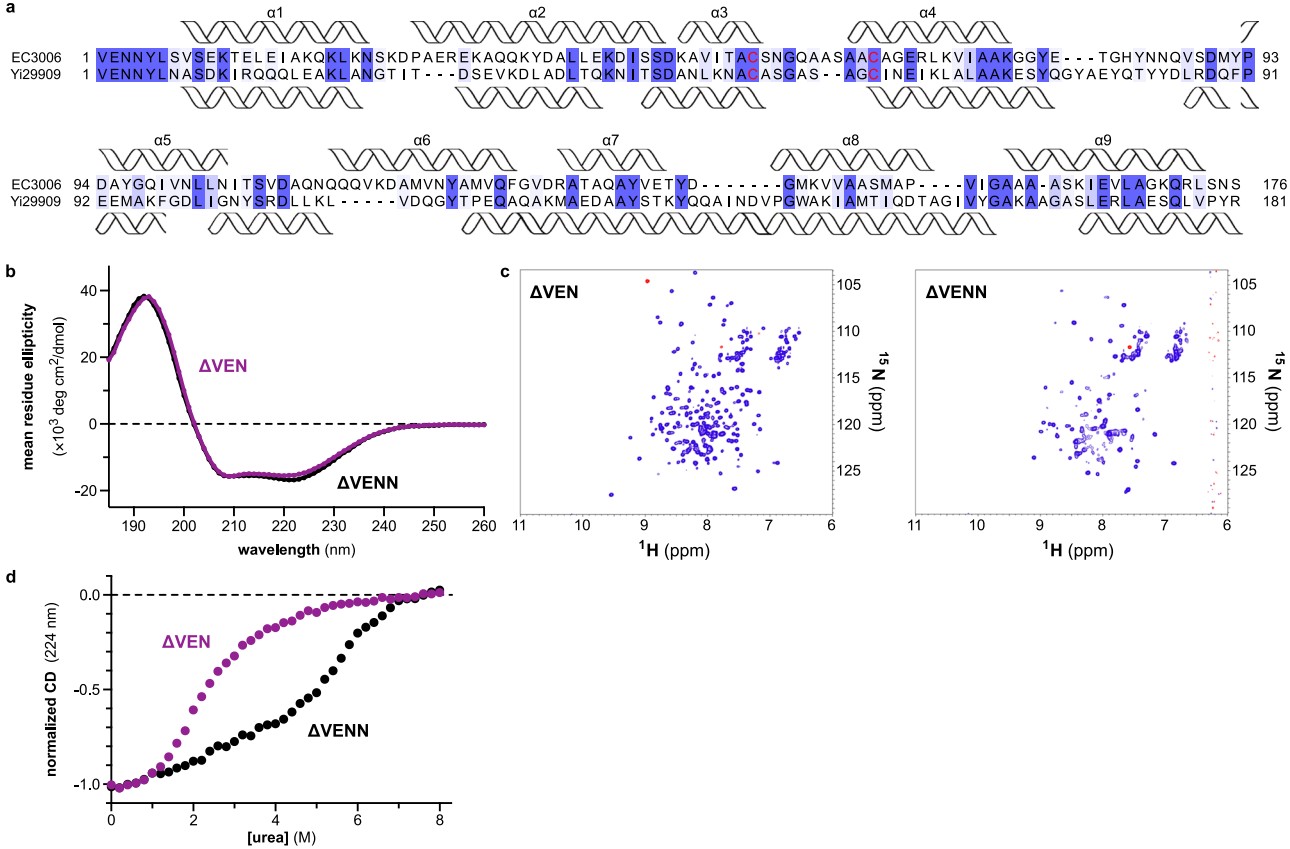

**Fig. 5 | The entry domain from CdiA-CT$^{Yi29909}$ undergoes a conformational switch. a** Alignment of entry domains from CdiA-CT$^{EC3006}$ and CdiA-CT$^{Yi29909}$. Secondary structure elements below the alignment correspond to predictions CdiA-CT$^{Yi29909}$. **b** CD spectra of ΔVEN and ΔVENN entry domains from CdiA-CT$^{Yi29909}$.

**c** $^1$H-$^{15}$N HSQC spectra of ΔVEN and ΔVENN entry domains from CdiA-CT$^{Yi29909}$. **d** Chemical denaturation profiles for ΔVEN and ΔVENN entry domains from CdiA-CT$^{Yi29909}$. The experiments in panels **b**–**d** were performed once. Source data are provided as a Source Data file.

have comparable thermodynamic stabilities. This conclusion is supported by urea melts of the isolated tRNase domain, which has the same unfolding energy as the ΔVEN entry domain (Supplementary Fig. 7e). The ΔVENN CdiA-CT$^{EC3006}$ appears to undergo at least two transitions during urea denaturation, consistent with the independent unfolding of the entry and tRNase domains (Supplementary Fig. 7e). In fact, arithmetic averaging of the isolated entry and tRNase domain datasets closely approximates the denaturation profiles of the corresponding CdiA-CT$^{EC3006}$ constructs (Supplementary Fig. 7f). Together, these results strongly suggest that the component domains of CdiA-CT$^{EC3006}$ are independent in structure and stability.

Given the striking influence of the N-terminus on entry domain structure, we tested whether Tyr5, which is predicted to be the N-terminal residue after processing, is required for the conformational switch. Tyr5 is not universally conserved across PtsG-dependent entry domains, but this position is usually an aromatic residue (Supplementary Fig. 8). CD spectroscopy shows that wild-type ΔVENN and ΔVENN-Tyr5Ala entry domains share very similar α-helical content (Fig. 4d), but the $^1$H-$^{15}$N HSQC spectrum of the ΔVENN-Tyr5Ala construct more closely resembles those of the ΔVEN and ΔVENNY variants (Fig. 4b and Supplementary Fig. 6e). The ΔVENN-Tyr5Ala entry domain also exhibits a cooperative unfolding transition at lower urea concentrations like the ΔVEN and ΔVENNY domains (Fig. 4c, e). Furthermore, the ΔVENN-Tyr5Ala version of the CdiA-CT$^{EC3006}$ does not enter polymyxin permeabilized cells in the outer-membrane bypass assay (Fig. 4f, lane 4), though this construct retains tRNase actvity in vitro (Fig. 4g, lane 4). These results show that the N-terminal residue of the processed entry domain controls a conformational switch that is critical for membrane translocation.

To test whether the switch occurs with other entry domains, we examined the CdiA-CT$^{Yi29909}$ from *Yersinia intermedia* ATCC 29909, which contains an uncharacterized entry domain linked to a toxin domain with predicted deaminase activity (Pfam:PF14424) (Supplementary Table 1). The CdiA-CT$^{Yi29909}$ and CdiA-CT$^{EC3006}$ entry domains only share ~23% sequence identity (excluding the VENN motif) but are predicted to have similar secondary structure content and a disulfide bond (Fig. 5a). CD spectroscopy shows that ΔVEN and ΔVENN versions of the CdiA-CT$^{Yi29909}$ entry domain are predominately α-helical (Fig. 5b). However, the ΔVEN domain has a $^1$H-$^{15}$N HSQC spectrum typical of a folded protein, whereas the ΔVENN domain has significantly fewer resonances (Fig. 5c). The ΔVEN and ΔVENN domains also show profound differences during chemical denaturation, with a linear unfolding trajectory for ΔVENN and a cooperative transition for ΔVEN (Fig. 5d). These results suggest that the entry domain from CdiA-CT$^{Yint29909}$ also undergoes a conformational change when cleaved from the effector.

### The entry domain is stabilized by a conserved disulfide that promotes translocation

Residues Cys55 and Cys65 of the PtsG-dependent entry domain form a disulfide that appears to stabilize the N-terminal subdomain (see Fig. 2a). To explore the functional significance of this linkage, we constructed a Cys-free version of the ΔVENN entry domain containing Cys55Ser and Cys65Ser substitutions. CD spectroscopy suggests that wild-type and Cys-free ΔVENN entry domains have similar α-helical content (Fig. 6a), and the $^1$H-$^{15}$N HSQC spectrum for the Cys-free ΔVENN domain is more similar to those of the ΔVEN and ΔVENNY entry domains (Fig. 6b). These data indicate that the entry domain retains

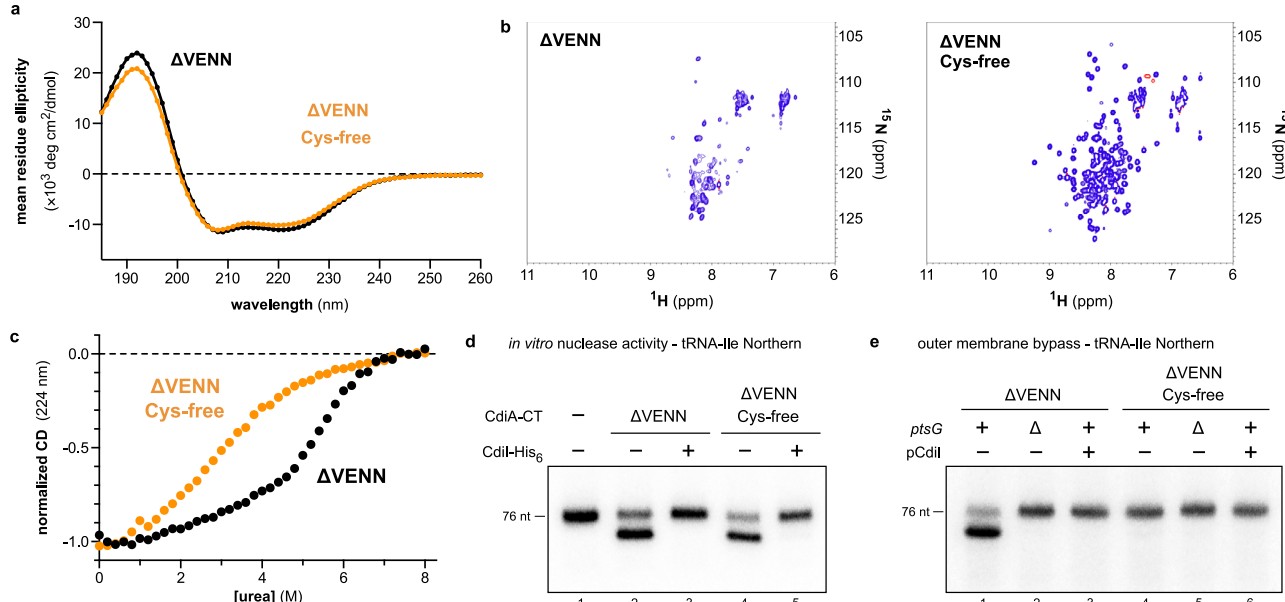

**Fig. 6 | The entry domain is stabilized by a conserved disulfide that promotes translocation. a** CD spectra of wild-type and Cys-free ΔVENN entry domains. **b** $^1$H-$^{15}$N HSQC NMR spectra of wild-type and Cys-free ΔVENN entry domains. **c** Chemical denaturation of wild-type and Cys-free ΔVENN entry domains. **d** In vitro nuclease activity. *E. coli* total RNA was treated with CdiA-CT$^{EC3006}$ proteins in the absence and presence of purified CdiI$^{EC3006}$ immunity protein (CdiI-His$_6$). **e** Purified CdiA-CT$^{EC3006}$

was incubated with polymyxin B (PMB) treated *E. coli* cells. Total RNA was isolated for Northern blot analysis using a probe to tRNA$^{Ile}$. The experiment in panel **e** was repeated independently three times with similar results. The experiments in panels **a** and **c** were repeated independently twice with similar results. The experiments in panels **b** and **d** were performed once. Source data are provided as a Source Data file.

significant structure in the absence of the disulfide bond. The Cys-free ΔVENN entry domain undergoes a markedly non-cooperative denaturation, though it unfolds at lower urea concentrations than wild-type (Fig. 6c). The Cys55Ser/Cys65Ser substitutions have no discernable effect on tRNase activity when introduced into the ΔVENN CdiA-CT$^{EC3006}$ construct (Fig. 6d, compare lanes 2 and 4), but the disulfide is important for translocation because Cys-free ΔVENN CdiA-CT$^{EC3006}$ does not enter polymyxin permeabilized cells (Fig. 6e, compare lanes 1 and 4).

## CdiA-CT$^{EC3006}$ translocation during cell-mediated delivery

We next tested whether structural perturbations to the entry domain affect tRNase delivery during cell-mediated CDI. We grafted wild-type, Asn4Ala (VENA), Tyr5Ala (VENNA), and Cys-free versions of the CdiA-CT$^{EC3006}$ onto full-length CdiA$^{STECO31}$, which delivers toxin upon recognition of Tsx receptors on target bacteria[2]. Inhibitor cells that deploy the wild-type CdiA$^{STECO31}$-CT$^{EC3006}$ chimera outcompete *tsx*$^+$ target cells ~10$^4$-fold after 1 h, whereas the CdiA-CT processing-defective VENA construct provides no competitive advantage (Supplementary Fig. 9a). Surprisingly, the Tyr5Ala and Cys-free variants confer the same growth advantage as wild-type CdiA-CT$^{EC3006}$ under these conditions (Supplementary Fig. 9a). We also attempted to generate an effector that delivers the ΔVEN version of CdiA-CT$^{EC3006}$ by inserting an additional Asn residue into the VENN motif (VENN↓NYL), and an effector that releases the ΔVENNY form by deleting residue Tyr5 (VENN↓L). These latter chimeras are inactive (Supplementary Fig. 9a), but immunoblot analysis revealed that the CT regions of both effectors are not processed efficiently in the presence of *tsx*$^+$ target bacteria (Supplementary Fig. 9b). Thus, the impact of an altered processing site cannot be evaluated under physiological conditions because the VENNN and VENNL variants do not release CdiA-CT fragments into the target-cell periplasm.

The efficacy of the Tyr5Ala and Cys-free constructs during cell-mediated CDI was unexpected, given that these mutations abrogate cell entry in outer-membrane bypass assays. We reasoned that entry defects could be masked during prolonged co-culture, in which inhibitor cells continually synthesize new CdiA effectors for multiple rounds of toxin

delivery. Therefore, we examined the kinetics of growth inhibition under conditions that preclude repeated toxin delivery. Inhibitor cells were treated with spectinomycin (Spm) to block protein synthesis, then mixed at a 1:1 ratio with Spm-resistant *E. coli tsx*$^+$ target bacteria. Under these conditions, target-cell viability decreased almost 100-fold after 30 min with wild-type inhibitors, then remained relatively constant over the following 30 min (Fig. 7a). By contrast, viable target bacteria increased ~5-fold in co-cultures with either mock (CDI$^-$) or VENA inhibitor cells (Fig. 7a). Northern blot analysis revealed significant tRNA$^{Ile}$ cleavage in the wild-type competition co-culture after 15 min (Fig. 7b, lane 7). This nuclease activity reflects CdiA-CT$^{EC3006}$ intoxication because cleaved tRNA$^{Ile}$ was not detected at any time during co-culture with Δ*tsx* mock target cells (Fig. 7b, lanes 6, 11, 16, and 21). Because inhibitor cells are immune to intoxication, all of the cleaved tRNA$^{Ile}$ is from the target-cell population. Thus, the in vivo nuclease activity correlates well with the loss of target-cell viability.

In Spm-supplemented media, the activities of the Tyr5Ala and Cys-free CdiA-CT$^{EC3006}$ variants are significantly attenuated. Target-cell viability decreased ~10-fold after 30 min with Tyr5Ala inhibitors, and only ~4-fold with inhibitors that deliver the Cys-free variant (Fig. 7a). In fact, after 30 min in the latter co-culture, target cells recommenced growth (Fig. 7a). Consistent with increased target-cell viability, tRNase activity was not detected in the Tyr5Ala and Cys-free co-cultures at 15 min (Fig. 7b, lanes 9 and 10), and only became apparent upon further incubation. In principle, this diminished potency could result from defective CdiA-CT processing. However, immunoblot analysis showed that the CT regions of the Tyr5Ala and Cys-free variants are processed with similar kinetics as the wild-type (Fig. 7c). Thus, the Tyr5Ala and Cys-free CdiA-CT$^{EC3006}$ variants are released into the target-cell periplasm, but translocate to the cytosol less efficiently than wild-type. Given that the Cys-free entry domain is less stable to urea denaturation than wild-type (see Fig. 6c), we wondered whether the disulfide bond protects the CdiA-CT$^{EC3006}$ from adventitious degradation in the periplasm. We incubated wild-type and Cys-free versions of ΔVENN CdiA-CT$^{EC3006}$ with *E. coli* cytoplasmic membrane vesicles and monitored degradation. Wild-type CdiA-CT$^{EC3006}$ is relatively stable over 40 min,

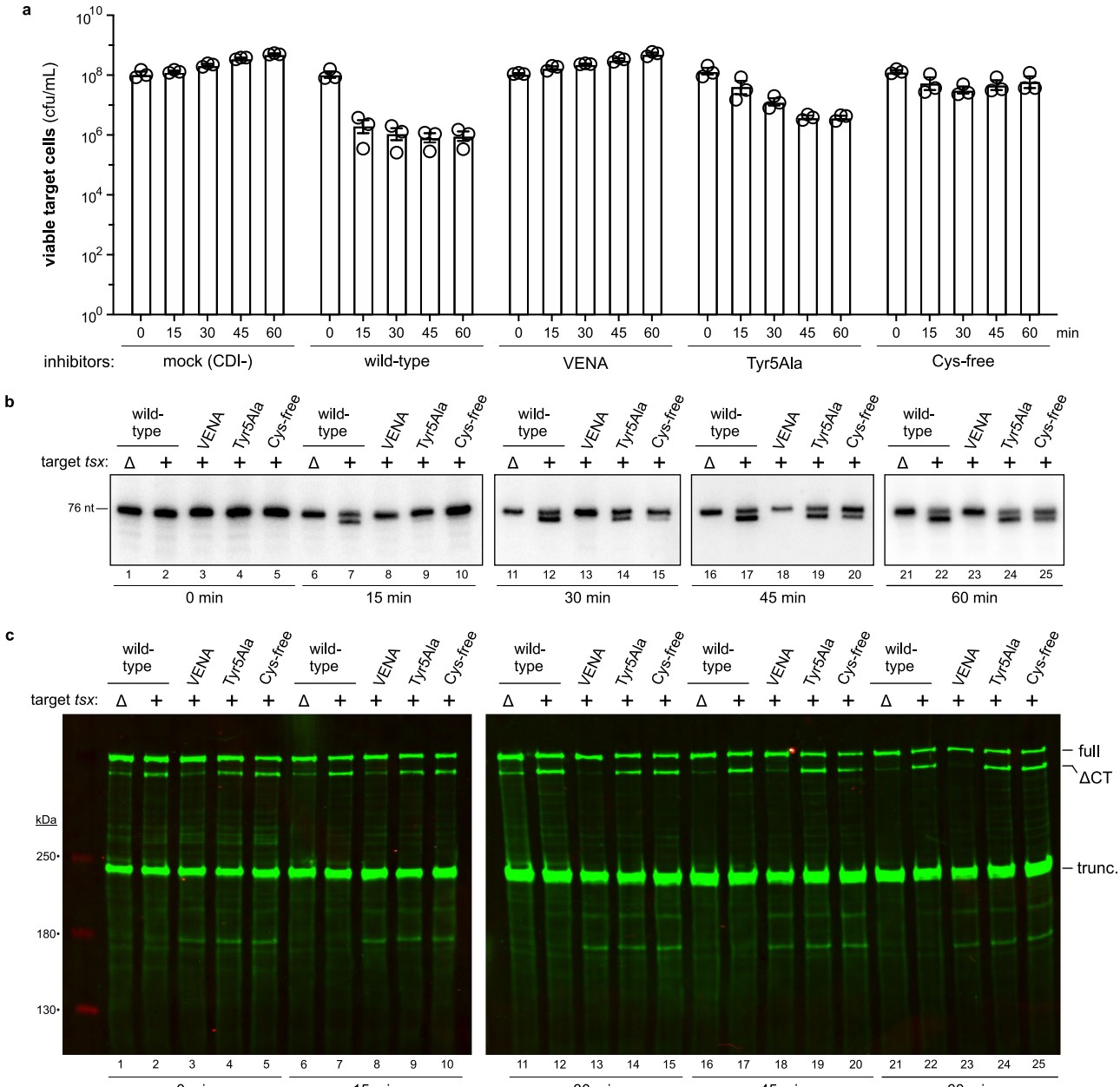

**Fig. 7 | CdiA-CT^EC3006 translocation during cell-mediated delivery. a** Competition co-cultures. Target bacteria were mixed at a 1:1 ratio with inhibitor cells that deploy the indicated CdiA-CT^EC3006 variants. Viable target cells were enumerated as colony-forming units (cfu) per mL every 15 min. Presented data were the average ± SEM for three independent experiments. **b** In vivo nuclease activity. Total RNA was isolated from the co-cultures in panel a for Northern blot analysis using a probe to tRNA^Ile. **c** CdiA-CT^EC3006 processing. Urea-soluble protein was isolated from the co-cultures in panel a for immunoblot analysis using polyclonal antisera to the N-terminal TPS transport domain of CdiA. Full-length, ΔCdiA-CT, and truncated CdiA proteins are indicated. The experiment in panel **a** was repeated independently three times with similar results. The experiments in panels **b**, **c** were performed independently twice with similar results. For panels **e**, **f**, denaturation of CdiA-CTs was repeated independently three times with similar results and tRNase domain denaturation was performed once. Source data are provided as a Source Data file.

but the Cys-free variant is cleaved progressively over time (Supplementary Fig. 10a). Reverse-phase HPLC and mass spectrometry analyses revealed that Cys-free CdiA-CT^EC3006 is cleaved between residues Lys71 and Val72 (Supplementary Fig. 10b, c). Although ΔVEN, ΔVENNY, and ΔVENN-Tyr5Ala entry domains are also less stable to urea denaturation, the corresponding CdiA-CT^EC3006 constructs are not cleaved when incubated with membrane vesicles (Supplementary Fig. 10d).

## CdiA-CT processing is independent of the proton gradient and inner-membrane receptors

CdiA-CT processing is thought to be mediated by the pretoxin-VENN domain[2], though it is unclear how auto-cleavage is triggered in the

target-cell periplasm. Given that integral membrane proteins are hijacked for transport to the cytosol, we tested whether these receptors are also required for CdiA-CT processing. Immunoblot analysis showed that the wild-type CdiA-CT^EC3006 is processed with similar efficiency when delivered into ΔptsG or ptsG^+ target cells (Fig. 8a, lanes 2 and 4). Although the CdiA-CT^EC3006 is processed with ΔptsG target cells, Northern blot analysis indicates that the released toxin does not enter the cytosol (Fig. 8b, compare lanes 2 and 4). Similar results were obtained with an effector that delivers the AcrB-dependent CdiA-CT^Ym43969 region. We note that this latter chimera shows significant Tsx-independent cleavage (Fig. 8a, lanes 5 and 7), which presumably reflects aberrant processing in the periplasm of inhibitor cells prior to target-

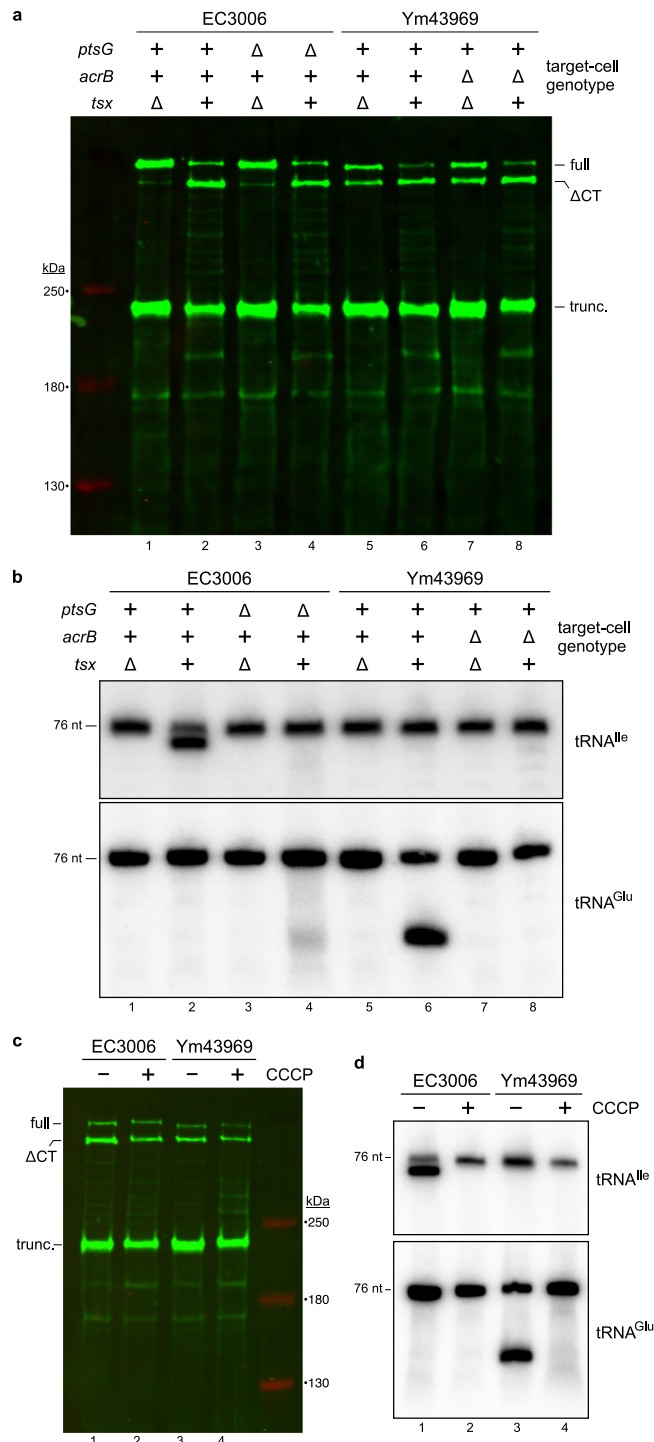

**Fig. 8 | CdiA-CT processing is independent of the proton gradient and inner-membrane receptors. a** CdiA-CT processing. Inhibitor cells that deploy CdiA-CT[EC3006] or CdiA-CT[Ym43969] were mixed at a 1:1 ratio with target bacteria of the indicated genotypes. Urea-soluble protein was isolated from co-cultures for immunoblot analysis using polyclonal antisera to the N-terminal TPS domain of CdiA. **b** In vivo nuclease activity. Total RNA was isolated from the co-cultures in panel **a** for Northern blot analysis using probes to tRNA[Ile] and tRNA[Glu]. **c** Inhibitor cells that deploy CdiA-CT[EC3006] or CdiA-CT[Ym43969] were mixed at a 1:1 ratio with target bacteria in the absence or presence of carbonyl cyanide *m*-chlorophenylhydrazone (CCCP). Urea-soluble protein was analyzed by immunoblotting with polyclonal antisera to the N-terminal TPS transport domain of CdiA. **d** In vivo nuclease activity. Total RNA was isolated from the co-cultures in panel **c** for Northern blot analysis using probes to tRNA[Ile] and tRNA[Glu]. The experiment in panel **a** was repeated independently three times with similar results. The experiments in panels **b**–**d** were performed twice independently with similar results. Source data are provided as a Source Data file.

## Discussion

CdiA-CT[EC3006] import is mediated by its N-terminal cytoplasm-entry domain, which exploits PtsG to promote translocation into the cytosol[18]. The mechanism of membrane translocation is unknown, but the entry domain has been proposed to bind PtsG to facilitate direct penetration into the lipid bilayer[18]. Crystallography reveals that the PtsG-dependent entry domain is composed of two α-helical sub-domains, though none of the helices is sufficiently hydrophobic to form a stable transmembrane segment. Additionally, the structure probably does not reflect the active conformation of the entry domain because the crystallized form carries the VENN peptide, which prevents cell entry in outer-membrane bypass assays. In fact, the ΔVENN form of CdiA-CT[EC3006] is the only proteolytic fragment that enters the cytoplasm efficiently. These observations indicate that precise cleavage after the VENN motif is required to convert the entry domain into a translocation competent state. The processed domain becomes significantly more resistant to chemical denaturation, which is somewhat unexpected given that it must presumably unfold during membrane translocation. The ΔVENN domain also appears to form oligomers at high micromolar concentrations. Because processed CdiA-CTs do not accumulate to these concentrations under physiological conditions, self-association per se is unlikely to contribute to translocation. However, the propensity to aggregate may reflect a change in hydrophobic exposure, which in turn could facilitate binding to the membrane or to PtsG. Remarkably, these biophysical changes do not occur when the N-terminal Tyr5 residue is substituted with Ala. In the crystal structure, the side chain of Tyr5 stacks onto the main chain at residue Asp48 (see Fig. 2c). This unusual tertiary contact occurs at the α2-α3 junction and appears to disrupt what would otherwise be a continuous helix. It remains to be determined whether this interaction is relevant to the structural switch or whether it merely reflects a conformation stabilized by the crystal lattice.

The entry domains of CdiA-CT[EC3006] and CdiA-CT[Yi29909] both undergo structural transitions when cleaved from the VENN motif, but previous work suggests that the MetI-dependent entry domain is dynamic in the absence of processing[33]. These observations suggest that conformational flexibility is important for membrane translocation, but also indicate that the switch described here is not a universal feature of CDI toxin delivery. CdiA proteins from the Enterobacteriaceae collectively contain at least 29 distinct entry domain families, most of which have yet to be characterized experimentally (Supplementary Table 1). In general, entry domains are predicted to be predominately α-helical, and most contain at least one pair of Cys residues. Though the disulfide in CdiA-CT[EC3006] is not required for membrane translocation, it clearly stabilizes the entry domain and may protect it from adventitious degradation in the target-cell periplasm. Entry domains within a given family often exhibit significant variation in sequence. For example, PtsG-dependent entry domains from *E. coli*

cell recognition. Nevertheless, CdiA-CT[Ym43969] cleavage increases noticeably in the presence of *tsx*⁺ target bacteria (Fig. 8a, compare lanes 5 and 6), and processing is equivalent between *acrB*⁺ and Δ*acrB* target cells (Fig. 8a, lanes 6 & 8). Again, tRNase activity was only detected in *acrB*⁺ targets (Fig. 8b, lanes 6 and 8), indicating that released CdiA-CT[Ym43969] cannot enter the cytoplasm of Δ*acrB* mutants. Finally, we examined CdiA-CT processing and delivery in co-cultures treated with CCCP to dissipate the proton gradient across the cytoplasmic membrane. CCCP has no discernable effect on CdiA-CT processing (Fig. 8c), but tRNase activity is not detected in the treated co-cultures (Fig. 8d, lanes 2 and 4). Thus, CdiA-CTs can be delivered into the periplasm of de-energized target cells, but the proton motive force (pmf) is required for subsequent translocation into the target-cell cytoplasm.

strains 3006, NC101, and STEC_O31 share ~70% pair-wise identity, with most of the polymorphism localized to helix α9 (Fig. 2b and Supplementary Fig. 8). Because helix α9 contacts the tRNase domain in the CdiA-CT[EC3006] crystal structure, it appears that each entry domain has evolved a unique interaction surface to accommodate its distinct toxin cargo. However, we find that the component domains of CdiA-CT[EC3006] retain their overall structural and thermodynamic properties when separated, suggesting that they are autonomous. In fact, the entry domain from CdiA-CT[EC3006] is able to guide a heterologous PD-(D/E)xK phosphodiesterase into target bacteria even though the latter nuclease is not associated with the PtsG-dependent entry domain in any naturally occurring CdiA protein[18]. The activity of this artificial CdiA-CT hybrid suggests that toxin domains are passive passengers during membrane translocation. Such modularity should allow the assembly of novel CdiA-CTs through genetic recombination, consistent with the observation that most entry domain families are associated with multiple toxin types (see Supplementary Fig. 8 and Supplementary Table 1).

CdiA effectors share common features with colicin toxins released by some *E. coli* isolates. Like CdiA, colicins also recognize specific cell-surface receptors and deliver functionally similar C-terminal nuclease domains into target bacteria[34]. Though analogous, CdiA effectors and colicins use very different strategies to deliver toxins across the cell envelope. All colicins exploit either the Tol or the Ton system to gain access to the target-cell periplasm. Tol and Ton are both multi-protein complexes with cytoplasmic membrane components that harness energy from the proton gradient to perform mechanical work in the outer membrane[35,36]. Colicins engage Tol/Ton by threading their intrinsically unstructured N-termini into the periplasm through the central lumina of the outer-membrane porins[37]. Once interactions with the Tol/Ton machinery are established, the colicin is imported to the periplasm through pmf-driven mechanical pulling forces[38]. By contrast, CDI toxin translocation across the target-cell outer membrane is independent of Tol/Ton and does not require the pmf[28]. The cytoplasm-entry mechanisms of colicins and CDI toxins also appear to be distinct. All nuclease colicins contain a pyocin S translocation domain (Pfam: PF06958), which is required for FtsH-dependent cytoplasmic import[39–41]. FtsH is a membrane-embedded AAA + metalloprotease that normally acts to degrade misfolded membrane proteins. FtsH is thought to use its ATPase activity to transport the toxin across the membrane and its protease activity to release the nuclease domain into the cytoplasm[40,42]. Intriguingly, pyocin S translocation domains are found in some type VI secretion system effectors that carry C-terminal HNH nuclease domains[43,44], suggesting that FtsH mediates import after deposition into the periplasm. We note that some CDI toxins also require FtsH to enter target bacteria[18] (Supplementary Table 1), but their entry domains are not related to the pyocin S translocation domain, and the import mechanisms are likely distinct. Given that CDI nucleases typically have low thermodynamic stability[45], they may risk complete degradation by FtsH if imported via the colicin pathway. This lability may have necessitated the evolution of alternative membrane translocation strategies for CDI toxins.

## Methods
### Bacterial strains and growth conditions
Bacterial strains are presented in Supplementary Table 2. Bacteria were cultured in M9 minimal medium, lysogeny broth (LB), or LB agar at 37 °C. Unless indicated otherwise, media were supplemented with antibiotics at the following concentrations: 150 μg/mL ampicillin (Amp), 100 μg/mL chloramphenicol (Cm), 50 μg/mL kanamycin (Kan), 200 μg/mL spectinomycin (Spc), and 25 μg/mL tetracycline (Tet). All deletion mutations were transferred from the Keio collection[46] of *E. coli* single-gene knockout strains using bacteriophage P1*vir* mediated general transduction. The Δ*ptsG::kan* allele was transferred into *E. coli*

MG1655 to generate strain CH12743. The Δ*ptsG::kan* mutation was also introduced into strain CH14016[2], with the kanamycin-resistance cassette subsequently removed using pCP20[47] encoded FLP recombinase to generate strain CH1672. The Δ*acrB::kan* allele was transferred into *E. coli* CH7367[2] to generate strain CH1738. The kanamycin-resistance cassette was removed from CH1738 using pCP20 to generate strain CH1739. The Δ*acrB::kan* mutation was also transferred into strain CH14016 and the kanamycin-resistance cassette was removed to generate strain CH1673.

### Plasmid constructions
All plasmids and oligonucleotide primers are presented in Supplementary Tables 3, 4, respectively. The coding sequences for CdiA-CT[EC3006] (corresponding to residues Val2921 – Lys3257 of EC3006_4140; Genbank: EKI34460.1) and CdiI[EC3006] (EC3006_4139; Genbank: EKI34459.1) were synthesized by Genscript (Piscataway, NJ) and supplied in the pUC57 vector (pCH6283). The *cdiA-CT/cdiI*[EC3006] module was PCR amplified with primers 209F46/209R50, then treated with T4 DNA polymerase and dTTP for insertion into plasmid MCSG58 using ligation-independent cloning procedures[48,49]. The resulting construct pMCSG58-APC200209 produces CdiA-CT[EC3006] carrying the N-terminal VENN motif and CdiI[EC3006] with a C-terminal His₆ affinity tag. Plasmid pMCSG63-APC200209, producing CdiI with the TEV protease-cleavable N-terminal His₆ tag has been described previously[25].

Plasmid pCH978 produces CdiA-CT[EC3006] (+VENN) and CdiI[EC3006]-His₆ as described previously[25]. pCH978 was amplified with primers CH4471/CH3245 (ΔVE), CH4470/3245 (ΔVEN), CH4436/CH3245 (ΔVENN), NB042/CH3245 (ΔVENNY), and NB045/CH3245 (ΔVEN-NYSLV) to introduce the TEV protease recognition sequence at various sites within and adjacent to the VENN motif of CdiA-CT[EC3006]. The resulting fragments were digested with KpnI/XhoI and ligated to pMCSG63 to generate plasmids pCH14274 (ΔVE), pCH14273 (ΔVEN), pCH14272 (ΔVENN), pCH14283 (ΔVENNY), and pCH14284 (ΔVEN-NYSLV). Similarly, the Tyr5Ala variant of the ΔVENN CdiA-CT[EC3006] construct was amplified with primers CH5112/CH3245 and ligated to pMCSG63 to generate pCH368 (ΔVENN-Y5A). Cys55Ser and Cys65Ser substitutions were made using overlap-extension PCR[50]. Plasmid pCH6283 was amplified with CH4436/CH3871 and CH4280/CH3245, and the two products were joined by amplification with primers CH4436/CH3245. The final product was digested with KpnI/XhoI and ligated to pMCSG63 to generate plasmid pCH14275 (ΔVENN Cys-free). The corresponding CED expression constructs were made in the same manner with primers CH4470/CH5199 (ΔVEN), CH4436/CH5199 (ΔVENN and ΔVENN-No Cys), NB042/CH5199 (ΔVENNY), and CH5112/CH5199 (ΔVENN-Y5A). These fragments were ligated to pMCSG63 via KpnI/XhoI to generate plasmids pCH391 (ΔVEN-CED), pCH392 (ΔVENN-CED), pCH396 (ΔVENNY-CED), pCH394 (ΔVENN-Y5A-CED) and pCH393 (ΔVENN-No Cys-CED). A fragment encoding the CdiA-CT[EC3006] tRNase domain and CdiI[EC3006] was amplified using CH5475/CH3245 and ligated to pMCSG63 via KpnI/XhoI to generate plasmid pCH789.

To append the ssrA(DAS) degron onto CdiI[Ym43969] for in vivo toxin activation, the *cdiA-CT/cdiI*[Ym43969] module was amplified from plasmid pCH12847[26] with primers CH3975/CH3977, and the product ligated to pCH7171[45] using NcoI/SpeI restriction sites to generate plasmid pCH1735. This latter plasmid was propagated in *E. coli* X90 Δ*clpX* Δ*clpA::kan* (CH7157). The *cdiA-CT/cdiI*[Ym43969] module was also amplified with CH4932/CH4933 and ligated to pMCSG63 via KpnI/XhoI to generate plasmid pCH465 (ΔVEDN). Plasmid pCH1737 that expresses the *cdiI*[Ym43969] immunity gene has been described[26]. The entry domain coding sequence from *Y. intermedia* ATCC 29909 was amplified with primer pairs CH5467/CH5378 (ΔVEN) and CH5355/CH5378 (ΔVENN), and the products ligated to pMCSG63 using KpnI/XhoI sites to generate plasmids pCH891 and pCH1188, respectively.

A NheI restriction site was introduced upstream of the VDNN coding sequence of *cdiA*[STECO31] to facilitate chimera construction. A fragment of *cdiA*[STECO31] was amplified with primers CH4282/CH4283 and ligated to pCH13658[2] using AflIII/XhoI restriction sites to generate plasmid pCH13709. Wild-type *cdiA-CT/cdiI*[EC3006] was amplified from pCH11483[18] with primers CH4262/CH3245 and ligated to pCH13709 via NheI/XhoI to generate plasmid pCH5019. The *cdiA-CT/cdiI*[Ym43969] module was amplified from pCH12847 with primers CH4262/CH4933 and ligated to pCH13709 via NheI/XhoI to generate plasmid pCH1681. The VENA, VENNA, VENNN, and VENNL variants were amplified from pCH13709 using reverse primer CH3245 in conjunction with CH5183 (VENA), CH5184 (VENNA), CH5298 (VENNN), and CH5299 (VENNL) forward primers. Products were reamplified with primers CH4262/CH3245 and ligated into pCH13709 via NheI/XhoI to generate plasmids pCH6587 (VENA), pCH6588 (VENNA), pCH523 (VENNN), and pCH524 (VENNL). Cys55Ser/Cys65Ser (Cys-free) substitutions were introduced into the *cdiA*[STECO31]-CT[EC3006] chimera using overlap-extension PCR with primer CH4262/CH3871 and CH4280/CH3245 to generate plasmid pCH6590.

### Protein expression, purification, and crystallization

*E. coli* BL21 (DE3) cells carrying pMCSG58-200209 were grown for 6 h at 37 °C in 2 mL of LB medium supplemented with 100 µg/mL Amp. The culture was diluted 1:100 into 50 mL of M9 minimal media supplemented with non-inhibitory amino acids, 5 g glucose, 0.5% (vol:vol) glycerol, 100 µg/mL Amp and trace minerals and vitamins, and incubated overnight at 37 °C. The overnight culture was diluted 1:100 into 1 L of the same M9 minimal media, and cells were grown in a 2 L polyethylene terephthalate beverage bottle to $OD_{600}$ -0.8 at 37 °C, then cooled to 18 °C. Selenomethionine (Se-Met) was added to 60 µg/mL together with the inhibitory amino acids (L-isoleucine, L-leucine, L-lysine, L-phenylalanine, L-threonine, and L-valine) to final concentrations of 100 µg/mL. Protein production was induced with 0.5 mM isopropyl-D-thiogalactopyranoside (IPTG) and the cells were incubated overnight at 18 °C. Cells were collected by centrifugation, then washed and resuspended in 50 mM Tris (pH 8.0), 500 mM NaCl, 10 mM 2-mercaptoethanol (2-ME), 10% glycerol and Complete Protease Inhibitor Cocktail (Roche Mannheim). Cells were lysed using FastBreak™ reagent (Promega) with 10 µg/mL of lysozyme and 500 U of Benzonase (Novagen EMD Millipore). The lysate was clarified by centrifugation and filtration before loading onto a Ni (II) Sepharose HisTrap column (GE Healthcare). Proteins were eluted in lysis buffer supplemented with 250 mM imidazole. Fractions containing the CdiA-CT•CdiI[EC3006] complex were pooled and run on a Hi Load 26/60 Superdex 200 size-exclusion column equilibrated with 20 mM Tris-HCl (pH 7.5), 150 mM NaCl, 2 mM dithiothreitol (DTT). The CdiA-CT•CdiI[EC3006] complex was concentrated to 10 mg/mL using an Amicon Ultracel 10 K concentrator.

In addition, the Se-Met labeled complex variant encoded by the pMCSG63-200209 construct was purified and concentrated to 10 mg/mL as described previously[25]. The CdiA-CT•CdiI[EC3006] complex preparations from both constructs, pMCSG58-200209 and pMCSG63-200209, were concentrated and crystallized in Crystal Quick 96-well plates (Greiner Bio-one) using a Mosquito nanoliter liquid handler (TTP Labtech) and commercially available crystallization screens. Protein and crystallization solution (400 nL) were aliquoted into wells at a 1:1 (vol:vol) ratio, with protein concentrations of 8 and 10 mg/mL, respectively. The plates were centrifuged to promote mixing and incubated a 4 °C.

### Data collection, structure solution, and refinement

The CdiA-CT•CdiI[EC3006] complex originating from vector pMCSG58-200209 crystallized from Pi-minimal Screen HTS in 34.3% PEG1000, 150 mM malate (pH 5.0), 70 mM sodium/potassium tartrate. The sample obtained from pMCSG63-200209 crystallized from the MIDAS screen condition with 30% glycerol ethoxylate, 0.2 M ammonium acetate, 0.1 M MES (pH 6.5). Prior to flash-cooling in liquid nitrogen, the crystals from the Pi-minimal screen were cryo-protected in crystallization liquor supplemented with 1% glycerol. The crystals from the MIDAS screen were frozen directly. Diffraction data were collected on the Structural Biology Center 19-ID beamline at the Advanced Photon Source, Argonne National Laboratory. Images were recorded on an ADSC Q315r detector at 100 K near the selenium K-absorption edge for Se-Met anomalous signal-based phasing. Images were processed using the HKL3000 suite[51], and intensities were converted to structure factor amplitudes using the Ctruncate program[52,53] from the CCP4 package[54]. Data collection and processing statistics are presented in Table 1.

The structure from the Pi-minimal screen was solved by molecular replacement using 6CP8 to model CdiI[EC3006] and the tRNase domain of CdiA-CT[EC3006], followed by MR-SAD in Phaser[55] and rebuilding in Buccaneer[56] and Coot[57]. The initial model was used to phase the MIDAS dataset by molecular replacement in Phaser[58]. Due to better quality of data, the latter crystal form was further improved by manual adjustments in Coot and crystallographic refinement in Phenix[59] with 6 TLS groups. The asymmetric unit contains one CdiA-CT•CdiI[EC3006] complex. The final model contains residues Glu2-Gly77 and Tyr92-Phe223 of CdiA-CT[EC3006] (chain A, numbered from Val1 of the VENN motif), residues Ser6-Pro158 of CdiI[EC3006] (chain I) and 109 water molecules. The model has been built with methionine residues as the anomalous signal in the dataset was negligible, indicating poor Se-Met incorporation during expression. Structure figures were prepared using UCSF Chimera software (https://www.cgl.ucsf.edu/chimera/).

### Entry domain and CdiA-CT purification for biochemical, CD, and NMR analyses

For NMR experiments, expression strains were grown in M9 media supplemented with 2 g/L D-glucose and 1 g/L $^{15}NH_4Cl$ (Cambridge Isotope Laboratories). For all other biochemical characterization, cells were cultured in LB media supplemented with 2 g/L D-glucose and 100 µg/mL ampicillin for 3 h at 37 °C. Cells were collected by centrifugation and washed twice with 20 mM Tris-HCl (pH 7.5), 150 mM NaCl to remove residual glucose, then resuspended in fresh LB media supplemented with 100 µg/mL ampicillin, 100 µM IPTG and incubated at 37 °C with shaking. After 3 h, cells were collected by centrifugation at $4420 \times g$ in a Beckman JA-10 rotor for 15 min and the cell pellets were frozen at –80 °C. Frozen cells were resuspended in ice-cold lysis buffer [50 mM sodium phosphate (pH 7.6), 300 mM NaCl, 20 mM imidazole, 200 µM DTT] and broken by passage through a French pressure cell. Lysates were clarified at $14,600 \times g$ in a Beckmann JA-20 rotor for 1 h at 4 °C, and the supernatants were applied to a $Ni^{2+}$-loaded Chelating Sepharose Fast Flow (GE Healthcare) column. The column was washed with six volumes of lysis buffer. Untagged CdiI[EC3006] was then eluted with 2.5 column volumes of 50 mM sodium phosphate (pH 7.6), 300 mM NaCl, 20 mM imidazole, 8 M urea, 200 µM DTT. $His_6$-tagged entry domain and CdiA-CT variants were eluted with lysis buffer supplemented with 250 mM imidazole. Eluates were dialyzed against 50 mM sodium phosphate (pH 7.6), 300 mM NaCl, 20 mM imidazole, and 500 µM DTT overnight at 4 °C. $His_6$ affinity tags were cleaved with Tobacco Etch Virus (TEV) protease, followed by dialysis against 50 mM sodium phosphate (pH 7.6), 300 mM NaCl, 20 mM imidazole, 0.3 mM DTT at room temperature for 2 h. The dialysate was applied to a second pre-chilled column of Chelating Sepharose Fast Flow (GE Healthcare) matrix pre-loaded with $Ni^{2+}$ ions to remove the released $His_6$ peptide and $His_6$-tagged TEV protease. The column flow-through was concentrated to 2.5 mL using an Amicon Ultra-15 10 kDa centrifugal concentrator and applied to a PD-10 desalting column (GE Healthcare). The protein sample was eluted into a stirring solution of 50 mM sodium phosphate (pH 6.5) and allowed to oxidize overnight at room temperature. Proteins were concentrated using an Amicon Ultra-4

10 kDa centrifugal concentrator. Disulfide formation was confirmed by differential migration by SDS-PAGE under reducing and non-reducing conditions. Purified proteins were quantified by absorbance at 280 nm using the following extinction coefficients: CdiA-CT$^{EC3006}$ (+VENN, ΔVE, ΔVEN, and ΔVENN), 23,505 cm$^{-1}$ M$^{-1}$; CdiA-CT$^{EC3006}$ (ΔVENNY and ΔVENN-Tyr5Ala), 22,015 cm$^{-1}$ M$^{-1}$; CdiA-CT$^{EC3006}$ (ΔVENN Cys-free), 23,380 cm$^{-1}$ M$^{-1}$; CdiA-CT$^{EC3006}$ entry domain (ΔVEN and ΔVENN), 13,535 cm$^{-1}$ M$^{-1}$; CdiA-CT$^{EC3006}$ entry domain (ΔVENNY and ΔVENN-Tyr5Ala), 12,045 cm$^{-1}$ M$^{-1}$; CdiA-CT$^{EC3006}$ entry domain (ΔVENN Cys-free), 13,410 cm$^{-1}$ M$^{-1}$; CdiA-CT$^{Yint29909}$ entry domain (ΔVEN and ΔVENN), 23,505 cm$^{-1}$ M$^{-1}$; and CdiA-CT$^{EC3006}$ tRNase domain, 9970 cm$^{-1}$ M$^{-1}$.

## Purification of CdiI$^{EC3006}$

*E. coli* CH2016 cells[60] carrying plasmid pCH12802 were grown in LB medium supplemented with 100 μg/mL ampicillin, and CdiI$^{EC3006}$-His$_6$ production was induced with 100 μM IPTG. After 3 h, cells were collected by centrifugation at $4420 \times g$ for 15 min in a Beckman JA-10 rotor and frozen at –80 °C. Cells were resuspended in ice-cold lysis buffer containing 2 μM leupeptin, 2 μM peptstatin, 100 μM phenylmethylsulfonyl fluoride (PMSF), 5 μM tosyl-L-lysyl chloromethane HCl, then broken by passage through a French pressure cell. Cell lysates were clarified by centrifugation at $14,600 \times g$ in a Beckmann JA-20 rotor for 1 h at 4 °C. The supernatant was adjusted to 5 mM MgCl$_2$ and 500 μg of DNase I was added to digest DNA on ice for 20 min. The lysate was applied to a Ni$^{2+}$-loaded Chelating Sepharose Fast Flow (GE Healthcare) column pre-equilibrated with lysis buffer. The column was then washed with six column volumes of lysis buffer. His$_6$-tagged CdiI$^{EC3006}$ was then eluted with lysis buffer supplemented with 250 mM imidazole. The eluate was concentrated to <2.5 mL using an Amicon Ultra-15 10 kDa molecular weight cut-off centrifugal concentrator and applied to a PD-10 desalting column (GE Healthcare) equilibrated with 50 mM NaPO$_4$ (pH 6.5). Purified CdiI$^{EC3006}$-His$_6$ was quantified by absorbance at 280 nm (21,890 cm$^{-1}$ M$^{-1}$).

## Circular dichroism (CD) spectroscopy

Far UV CD spectra were acquired with protein at 5 μM in 20 mM sodium phosphate (pH 6.5) using a 0.1 cm path-length quartz cuvette. Near UV CD spectra were acquired with entry domains at 26–36 μM in 20 mM sodium phosphate (pH 6.5) using a 1.0 cm path-length quartz cuvette. Urea denaturation CD measurements were collected at 224 nm using a 1 cm path-length quartz cuvette with proteins at 400 nM (full-length CdiA-CT$^{EC3006}$) or 800 nm (cytoplasm-entry and tRNase domain) in 20 mM sodium phosphate (pH 6.5). The presented data were the average of two or three independent experiments. Thermodynamic properties were derived from the chemical denaturation data by the linear extrapolation method with curve fitting implemented in the Mathematica software suite (version 12.3, Wolfram Research)[61].

## NMR spectroscopy

$^1$H-$^{15}$N HSQC NMR experiments were performed with a Varian Inova 600 MHz spectrometer at 25 °C. Proteins were analyzed in 50 mM sodium phosphate (pH 6.5), 3 mM NaN$_3$. Spectra were acquired with CdiA-CT$^{EC3006}$ entry domains at the following concentrations: ΔVENN, 20 and 230 μM; ΔVEN, 302 μM; ΔVENNY, 135 μM; ΔVENN-Y5A, 303 μM; and ΔVENN Cys-free, 174 μM. ΔVENN and ΔVEN versions of the full CdiA-CT$^{EC3006}$ were analyzed at 124 and 103 μM (respectively) and isolated tRNase domain was analyzed at 197 μM. ΔVENN and ΔVEN entry domains from CdiA-CT$^{Yi29909}$ were analyzed at 115 and 99 μM, respectively. NMR data were processed with the NMRPipe software package[62].

## ANS fluorescence

Hen egg lysozyme and CdiA-CT$^{EC3006}$ entry domain variants (6 μM final protein concentration) were incubated with 50 μM 8-anilino-1-naphthalenesulfonic acid (ANS) in 50 mM sodium phosphate (pH 6.5) at ambient temperature. Samples were excited at 385 nm and fluorescence emission spectra were recorded from 400–600 nm using a Varian Cary Eclipse Fluorimeter.

## Size-exclusion chromatography−multi-angle light scattering (SEC-MALS)

Purified proteins were resolved on a Tosoh TSKgel G3000SWXL size-exclusion column in 50 mM sodium phosphate (pH 6.76) using an Agilent 1200 series high-performance liquid chromatography system. UV absorbance was monitored using an Agilent G7115A diode array, and the refractive index was measured using a Wyatt Optilab differential refractometer. Light scattering of the column eluates was monitored using a Wyatt DAWN 18-angle detector. Molecular mass and hydrodynamic radius calculations were performed using ASTRA 8.1 software.

## Outer-membrane bypass assays

*E. coli* strains were grown overnight at 37 °C with shaking in LB media supplemented with the appropriate antibiotics. Cells were adjusted to OD$_{600}$ = 0.1 in fresh LB media and incubated at 37 °C with shaking for 1 h 50 min to OD$_{600}$ ~0.8. Culture aliquots (2 mL) were centrifuged and the cells were resuspended at OD$_{600}$ = 16 in fresh pre-warmed (37 °C) LB media supplemented with 100 μg/mL polymyxin B sulfate and 1 μM purified CdiA-CT variants. Where indicated, carbonyl cyanide 3-chlorophenylhydrazone (CCCP) was included at 100 μM. Cell suspensions were incubated at 37 °C for 30 min, then collected by centrifugation, washed twice with fresh LB media, and processed for RNA extraction.

## In vitro nuclease assays

In vitro nuclease assays were performed in reaction buffer [20 mM Tris-HCl (pH 7.5), 100 mM NaCl, 5 mM MgCl$_2$, 10 mM 2-ME, and 100 μg/μL bovine serum albumin] with CdiA-CTs used at 1 μM final concentration. Where indicated, CdiI$^{EC3006}$-His$_6$ was included at 3 μM final concentration. Substrate tRNAs were first deacylated in 50 mM Tris-HCl (pH 8.9) for 1 h at 37 °C. Protein mixtures were equilibrated for 30 min at room temperature. Reactions were then initiated by the addition of *E. coli* total RNA to a final concentration of 800 ng/μL, followed by incubation for 1 h at 37 °C. Reactions were quenched with an equal volume of 25 SDS-urea gel loading buffer and run on 50% urea, 7.5% polyacrylamide gels buffered with 0.55 Tris-borate-EDTA. Gels were electroblotted to nylon membranes for hybridization with radiolabeled oligonucleotide probes as described above.

## In vivo toxicity and RNase assays

*E. coli* MG1655 *acrB*$^+$ (CH7286) and Δ*acrB* (CH1738) cells carrying plasmid pCH1735 were grown in LB media supplemented with Tet and 0.1% D-glucose for 150 min (OD$_{600}$ ~ 0.3). Cultures were split in two, and one was induced with 0.2% L-arabinose, while the other was supplemented with 0.4% D-glucose. Cultures were incubated for 3 h at 37 °C with shaking and cell growth was monitored by OD$_{600}$. Samples of each culture were harvested into an equal volume of ice-cold methanol 90 min after sugar supplementation. Cells were collected by centrifugation and frozen at –80 °C prior to RNA extraction.

## RNA isolation and analyses

Cell pellets were resuspended in guanidinium isothiocyanate (GITC)-phenol and total RNA was extracted as described in ref. 60. RNAs (6 μg) were resolved on 50% urea and 6% polyacrylamide gels buffered with 15 Tris-borate EDTA for 18 min at 350 V (constant). Gels were electroblotted to positively charged nylon membranes for Northern blot analysis. Blots were hybridized with [$^{32}$P]-labeled oligonucleotide probes specific for *E. coli* tRNA$_{GAU}^{Ile}$ (CH577) and tRNA$_{UUC}^{Glu}$ (CH1417)

(Supplementary Table 4), which have been validated by two-dimensional polyacrylamide gel electrophoresis[63] and previous Northern blot analyses[25,26]. Hybridized blots were visualized by phosphorimaging using Bio-Rad Quantity One software (version 4.5). All uncropped and unprocessed phosphorimager scans are provided in the Source Data files.

## Transposon mutagenesis

*E. coli* MFD *pir*⁺ cells carrying plasmid pSC189 were used as donors to introduce the *mariner* transposon into *E. coli* MG1655 cells by conjugation[64,65]. Donors and recipients were grown to mid-log phase in LB media supplemented with 30 μM diaminopimelic acid, then mixed and plated onto LB agar at 37 °C for 5 h. Cells from six independent matings were plated onto Kan-supplemented LB agar to select for transposon mutants. Each mutant pool was harvested into 1 mL 1× M9 salts and co-cultured with *E. coli* EPI100 inhibitor cells that produce chimeric CdiA^{EC93}-CT^{Ym43696} from plasmid pCH12847[26] to select for CDI^R clones. Surviving target bacteria were recovered on Kan-supplemented LB agar and subjected to two additional cycles of CDI^R selection. CDI^R clones were picked randomly from each independent mutant pool, and chromosomal DNA was isolated to identify transposon insertion sites. DNA was digested with NspI overnight at 37 °C followed by enzyme inactivation at 65 °C for 20 min. Digests were supplemented with 1 mM ATP and T4 DNA ligase and incubated overnight at 16 °C. The reactions were electroporated into *E. coli* DH5α *pir*⁺ cells and transformants selected on Kan-supplemented LB agar. The isolated plasmids were sequenced using oligonucleotide CH2260 to identify the junctions between the *mariner* transposon and genomic DNA.

## Competition co-cultures

*E. coli* EPI100 inhibitor cells carrying pCH12847 were mixed at a 1:1 ratio with *E. coli* MG1655 Δ*wzb* Δ*acrB* (CH1739) target bacteria in LB media and incubated at 37 °C with shaking. Where indicated, the target cells harbored plasmid pTrc99A (*cdiI*⁻), pCH1737 (*cdiI*^{Ym43696})[26], pZS21, or pCH1741 (pZS21-acrB)[66]. Viable target bacteria were enumerated as colony-forming units (cfu) per mL at 0 and 3 h of co-culture.

For the CdiA-CT^{EC3006} competitions shown in Supplementary Fig. 7a, *E. coli* MG1655 Δ*wzb* Δ*tsx* (CH14016) inhibitor cells carrying plasmids pET21b (CDI⁻), pCH5019 (wild-type), pCH6587 (Asn4Ala), pCH6588 (Tyr5Ala), pCH6590 (No Cys), pCH523 (VENNN), and pCH524 (VENNL) were co-cultured at a 1:1 ratio with MG1655 Δ*wzb::kan* (CH7286) target cells. Target-cell fitness was expressed as the competitive index, which is calculated as the final ratio of target to inhibitor cells divided by the ratio at $t = 0$. Competitive indices from three independent experiments are reported together with the average ± SEM.

## Kinetics of target-cell killing and CdiA-CT^{EC3006} toxin delivery

*E. coli* MG1655 Δ*wzb* Δ*tsx* cells (CH14016) carrying plasmids pET21b (CDI⁻), pCH5019 (wild-type), pCH6587 (Asn4Ala), pCH6588 (Tyr5Ala), and pCH6590 (No Cys) were diluted to OD_{600} ~0.05 in LB medium supplemented with Amp and cultured with shaking at 37 °C. Once in the mid-log phase, inhibitor cells were treated with Spc for 20 min to block protein synthesis, then mixed at a 1:1 ratio with Spc-resistant *E. coli* MG1655 Δ*wzb ara::spc* (DL8705) target cells in Spc-supplemented LB media. Co-culture samples were taken every 15 min and plated onto Spc-supplemented LB agar to enumerate viable target bacteria as cfu/mL. Samples were also taken to isolate RNA and protein for Northern blot and immunoblot analyses, respectively.

## SDS-PAGE and immunoblot analysis

Cells were collected by centrifugation and frozen at −80 °C. Frozen cells were resuspended in urea-lysis buffer [50% urea, 150 mM NaCl, 20 mM Tris-HCl (pH 8.0)] and subjected a freeze-thaw cycle to extract proteins. Urea-soluble proteins were resolved by SDS-PAGE on Tris-tricine 6% polyacrylamide gels run at 100 V (constant) for 3 h. Gels were soaked for 15 min in 25 mM Tris, 192 mM glycine (pH 8.6), 10% methanol, then electroblotted to low-fluorescence PVDF membranes using a semi-dry transfer apparatus at 17 V (constant) for 1 h. Membranes were blocked with 4% non-fat milk in 1× PBS for 1 h at room temperature and incubated with primary antibodies in 0.1% non-fat milk, 1× PBS overnight at 4 °C. Rabbit polyclonal antisera to the TPS domain of CdiA (residues Val33 - Gly285) were generated by Cocalico Biologicals Inc. (Reamstown, PA) as described by ref. [67]. Anti-TPS antisera were used at a 1:10,000 dilution to detect CdiA proteins by immunoblotting. Blots were incubated with 800CW-conjugated goat anti-rabbit IgG (1:40,000 dilution, LI-COR) in 0.1% non-fat milk in PBS. Immunoblots were visualized with an LI-COR Odyssey infrared imager. All uncropped and unprocessed infrared imager scans are provided in the Source Data files. Samples of the polyclonal anti-CdiA antisera are available from the corresponding author upon request.

## Isolation of cytoplasmic membrane vesicles

*E. coli* X90 cells were grown overnight at 37 °C with shaking in 30 mL of LB medium supplemented with rifampicin. The culture was inoculated into 700 mL of rifampicin-supplemented LB medium and grown for 5.5 h at 37 °C with shaking. Cells were collected by centrifugation for 15 min at 4420 × *g* in a Beckman JA-10 rotor and the pellets were frozen at −80 °C. Cells were suspended in 40 mL of ice-cold 10 mM Tris-HCl (pH 7.5), 34% (w/v) sucrose, 2 mM DTT. Lysozyme was added to 75 μg/mL, and the suspension was incubated on ice for 20 min. Cells were then gently stirred in an ice-chilled beaker (1 L) on a magnetic stir plate. In a dropwise fashion, 50 mL of 1 mM 1,10-phenanthroline, 1 mM PMSF, 5 mM EDTA, 1 mM DTT was added to the stirring cell suspension. Cell lysis was induced with the rapid addition of 400 mL of ice-cold 1 mM DTT. The osmolysed suspension was supplemented with 4 mM MgCl₂ and treated with 2 mg DNase I on ice for 20 min. Membranes were isolated by centrifugation at 105,000 × *g* in a Beckman 45 Ti rotor for 30 min at 4 °C. During centrifugation, a step gradient of 69, 61, 54, 47, and 40% (w/v) sucrose in 10 mM Tris-HCl (pH 7.5), 4 mM DTT was prepared in thin-wall ultracentrifuge tubes. Membrane pellets were suspended in ice-cold 34% (w/v) sucrose, 10 mM Tris-HCl (pH 7.5), 4 mM DTT and layered on top of the step gradient. Gradients were centrifuged at 113,000 × *g* in a Beckman SW-28 rotor overnight (~16 h) at 4 °C. The cytoplasm membrane fraction was isolated, diluted into 50 mM sodium phosphate (pH 6.5), and centrifuged at 186,000 × *g* in a Beckman 45 Ti rotor for 1 h at 4 °C. Cytoplasmic membranes were suspended in ice-cold 50 mM sodium phosphate (pH 6.5) by multiple passes through 16- and then 20-gauge syringe needles and stored at −80 °C. CdiA-CT^{EC3006} variants (3 μM) were incubated with membrane vesicles for up to 40 min in 50 mM sodium phosphate (pH 5.5) at ambient temperature.

## Statistics and reproducibility

Standard errors and standard deviations were calculated using either GraphPad Prism (version 9.4.0) or Microsoft Excel (version 16.63.1).

## Reporting summary

Further information on research design is available in the Nature Research Reporting Summary linked to this article.

# Data availability

The data generated during this study are provided within the manuscript or the Supplementary Information files. Structure datasets are available in the Protein Data Bank under accession codes 6CP8 and 6VEK. Sequences for CdiA^{EC3006} and CdiI^{EC3006} are available from Genbank under accession codes EKI34460.1 and EKI34459.1, respectively. Free induction decay (FID) NMR data have been deposited at the

Biological Magnetic Resonance Data Bank [https://bmrb.io/] under accession code 51540. Source data are provided with this paper.

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

## Acknowledgements

We thank Martin Kurnik at Wyatt Technologies (Santa Barbara, CA, USA) for SEC-MALS analyses, and Jeff Chen, Jaime Guerrero, Nasrat Hamid, Arslan Harmon, Calvin Lin, Megan Tran, Austin Ung, Sufia Hasan, and Nalin Zadoo for assistance with molecular cloning and protein purifications. This work was supported by National Institutes of Health grants GM117930 (C.S.H.), AI121789 (C.S.H and F.W.D), GM102318 (C.W.G., C.S.H., and subcontract to A.J.), GM115586 (A.J.), HHSN272201700060C (A.J.), and the U.S. Department of Energy, Office of Biological and Environmental Research, under contract DE-AC02-06CH11357 (A.J.). N.L.B was supported in part by a fellowship from the Santa Barbara Foundation Tri-Counties Blood Bank.

## Author contributions

Conceptualization, N.L.B., F.W.D., and C.S.H.; Methodology, N.L.B., V.J.P., K.M., K.S., D.Q.N, H.Z., L.M.S., and W.H.E.; Validation, V.J.P., H.Z., N.G.W., J.S.B., R.C., Z.N., Y.G., I.P.-H., and E.C.S.; Investigation, N.L.B., V.J.P., K.M., K.S., D.Q.N., B.J.C., and C.S.H.; Writing—original draft, N.L.B., K.M. and C.S.H.; Writing—review and editing, K.M., C.W.G., D.A.L., A.J., and C.S.H.; Visualization, N.L.B., K.M., H.Z., and C.S.H.; Funding acquisition, C.W.G., A.J., D.A.L., F.W.D., and C.S.H.; Supervision, A.J., F.W.D., and C.S.H.

## Competing interests

The authors declare no competing interests.
