## [Peer Review File · Nature Communications]

REVIEWERS' COMMENTS

Reviewer #1 (Remarks to the Author):

This is an impressive contribution. The striking dependence on the N-terminal sequence to initiate transport combined with the odd behavior of the structural properties of the domain make for a noteworthy story. This is perhaps the first clear documentation of the structural fluidity that has been postulated to be required for efficient membrane transport. The work is carefully documented and rigorously described and carefully interpreted. My critical comments focus on the term "molten globule" and what it really means here. CD indicates the presence of near native alpha helical structure while NMR indicates loss of stable structure. Denaturant melts indicate a loss of cooperativity. All this points to something akin to a molten globule, I agree. But what kind? For example, apocytochrome b562 (NSB 1,32) has the thermodynamic signature (low dCp for unfolding; not done here and DSC should be done) consistent with a "wet" MG. Versus the "dry" MG (Proteins 78, 2725). The NMR spectra were collected at 25 C. Structured states of these types of weakly stable/cooperative systems are often populated at lower temperature. What is the spectrum like at 10C? Finally, how intrinsically helical is the sequence i.e. can one imagine "helices on a string," which is decidedly not MG-like. In summary, I think this is an excellent contribution, but if "molten globule" is to be the highlight then a little more needs to be done to characterize that putative state.

Reviewer #2 (Remarks to the Author):

The "Proteolytic processing induces a molten-globule switch required for antibacterial toxin delivery" manuscript describes conformational changes in the E.coli cytoplasm-entry domain of CdiA in response to proteolytic processing. This is an excellent work that shows how structural information can be used effectively to reveal complex transmembrane protein translocation mechanisms. I only have several minor issues listed below:

Minor:

1. You offer some parallels and highlight differences between CdiA, anthrax and diphtheria toxin, and colicins. I wonder if in course of CdiA translocation, there is any ion-channel facilitated transport or membrane permeabilization takes place. Thus, in lines 372-373 you state the following: "Like CdiA, colicins also recognize specific cell-surface receptors and deliver C-terminal pore-forming and nuclease domains into target bacteria." Based on this, it sounds that there is also pore-formation involved, but I could not find it being mentioned anywhere else.
2. Lines 35-36. I had difficulties to understand this sentence, specifically this phrase: "...that contain one N-terminal 36 residue too many, or too few,...". Should it be a comma before "too many"? Even with that, I am not sure, I understand the meaning.
3. Please explain why polymyxin B E. coli cell permeabilization is needed.
4. Figs. 3A and 6A. What is the reason you chose presenting SEM instead of SDs?

Reviewer #3 (Remarks to the Author):

The manuscript by Bartelli et al. elucidates an important aspect of CdiA toxin delivery during the antagonistic process of contact-dependent growth inhibition (CDI) by bacteria. During CDI, the protein CdiA delivers its C-terminal toxin domain across the outer and inner membranes of a recipient bacterium. This process is known to occur in a specific stepwise manner that requires (1) specific protein domains within CdiA, (2) specific membrane proteins on the recipient cell, and (3) proteolytic processing of the C-terminal region (CT) prior to inner membrane translocation. In this manuscript, the authors elegantly describe how this third point, proper proteolytic processing of the CT, shifts a previously stable cytoplasmic entry domain into a dynamic molten globule structure. This specific processing is required for delivery of the toxin across the inner membrane. Furthermore, their data

supports earlier observations that this translocation step requires specific inner membrane protein receptors in the recipient cell and an intact proton gradient. The authors also present strong data suggesting that proteolytic processing of the CT occurs in the periplasm of the recipient bacterium and that this processing step is independent of the inner membrane receptors or proton gradient.

Throughout the manuscript, the authors use a robust series of experiments to analyze how CT processing, folding and translocation impact toxin delivery and ultimate growth inhibition. They are very thorough in their controls and their experiments were thoughtfully designed and elegantly performed. Although I do not have much experience in NMR spectroscopy, their explanation of the approaches and interpretations of the results were very clear. Overall, I found the manuscript to be extremely well-written and enjoyed reviewing the document. It is my opinion that this work is highly significant to the field of CDI biology. There is very little understanding of the CdiA transport processes, and this work provides valuable mechanistic insight into how this occurs. I did not observe any major flaws in their data analysis, interpretation, or conclusions. I applaud the authors for such fine work.

Reviewer #1 (Remarks to the Author):

This is an impressive contribution. The striking dependence on the N-terminal sequence to initiate transport combined with the odd behavior of the structural properties of the domain make for a noteworthy story. This is perhaps the first clear documentation of the structural fluidity that has been postulated to be required for efficient membrane transport. The work is carefully documented and rigorously described and carefully interpreted. My critical comments focus on the term "molten globule" and what it really means here. CD indicates the presence of near native alpha helical structure while NMR indicates loss of stable structure. Denaturant melts indicate a loss of cooperativity. All this points to something akin to a molten globule, I agree. But what kind? For example, apocytochrome b562 (NSB 1,32) has the thermodynamic signature (low dC_p for unfolding; not done here and DSC should be done) consistent with a "wet" MG. Versus the "dry" MG (Proteins 78, 2725). The NMR spectra were collected at 25 C. Structured states of these types of weakly stable/cooperative systems are often populated at lower temperature. What is the spectrum like at 10C? Finally, how intrinsically helical is the sequence i.e. can one imagine "helices on a string," which is decidedly not MG-like. In summary, I think this is an excellent contribution, but if "molten globule" is to be the highlight then a little more needs to be done to characterize that putative state.

We agree with the Reviewer's criticism and performed a series of new experiments to test whether the processed entry domain does indeed form a molten globule. An NMR spectrum of the Δ VENN domain acquired at 10 °C is indistinguishable from the original collected at 25 °C, arguing against a molten globule form. We also examined thermal unfolding of the Δ VENN and Δ VEN domains using a (temporarily) resurrected scanning calorimeter. We encounter several technical issues with this old instrument and are not confident in the calculated ΔC_p values. Ultimately, we decided to abandon the DSC work because other new experiments failed to support the molten globule model. For example, molten globules typically do not produce circular dichroism signals at near UV wavelengths due to the unstructured hydrophobic cores. However, we found that the Δ VENN domain actually exhibits greater negative ellipticity in the near UV than the other domain variants. In addition, the Δ VENN domain does not bind particularly well to ANS, which is a fluorescent dye commonly used to probe the molten globule state. Together, these results (presented as Supplementary Fig. 4 in the revision) have convinced us that processing does not induce a molten globule transition.

The most obvious explanation for NMR resonance broadening is protein aggregation; and we had already examined this possibility prior to the initial submission. The initial size-exclusion chromatography (SEC) results showed no evidence of higher order oligomerization for any of the entry domain variants. Nevertheless, in light of more recent results outlined above, we reexamined this issue more rigorously using multi-angle light scattering (MALS) to determine molecular mass unambiguously. These new experiments are presented in the revision (new Supplementary Fig. 5). Briefly, SEC-MALS indicates that Δ VEN and Δ VENN entry domains are primarily monomeric (at least after dilution during chromatography), but the Δ VENN variant elutes earlier than Δ VEN. Moreover, Δ VENN elution time is influenced by the amount loaded onto the size-exclusion column. Collectively, these results suggest that the Δ VENN domain self-associates at high micromolar concentrations, but the interactions are weak and reversible, allowing aggregated forms to dissociate as the sample is diluted during chromatography. Importantly, MALS shows that the Δ VENN domain is primarily monomeric at concentrations up to 20 μ M, and therefore oligomerization should have no effect on the cell-entry and urea denaturation experiments, which were conducted at 1 μ M and 0.8 μ M, respectively. Given the MALS data, we acquired another ^1H - ^{15}N HSQC spectrum of the Δ VENN domain at 20 μ M, which shows better dispersion than the high-concentration spectrum. These "new" peaks can also be seen in the original 230 μ M spectrum when rendered at a very low contour (Supplementary Fig. 6b), suggesting that the monomeric form is present at low levels. Comparisons of the various ^1H - ^{15}N HSQC spectra at low contour shows a handful of resonances that are unique to Δ VENN, but given the presumed heterogenous state of the domain at high concentration, we cannot make any firm conclusions concerning structural details.

To summarize, the Δ VENN form of the entry domain has interesting biophysical features that are quite distinct from other variants, but these differences are not due to molten globule formation. Accordingly, we have changed the manuscript title and significantly revised the Abstract, Introduction, Results and Discussion sections to conform to the new data.

Reviewer #2 (Remarks to the Author):

The “Proteolytic processing induces a molten-globule switch required for antibacterial toxin delivery” manuscript describes conformational changes in the E.coli cytoplasm-entry domain of CdiA in response to proteolytic processing. This is an excellent work that shows how structural information can be used effectively to reveal complex transmembrane protein translocation mechanisms. I only have several minor issues listed below:

Minor:

1. You offer some parallels and highlight differences between CdiA, anthrax and diphtheria toxin, and colicins. I wonder if in course of CdiA translocation, there is any ion-channel facilitated transport or membrane permeabilization takes place. Thus, in lines 372-373 you state the following: “Like CdiA, colicins also recognize specific cell-surface receptors and deliver C-terminal pore-forming and nuclease domains into target bacteria.” Based on this, it sounds that there is also pore-formation involved, but I could not find it being mentioned anywhere else.

We see the Reviewer's point and apologize for the confusion. We meant only to indicate that colicins and CdiA proteins deliver a similar variety of toxin domains – chiefly RNases, DNases and ionophores – not to imply that pore-formation contributes to CDI toxin delivery. We have removed the reference to pore-forming domains, and the remainder of the paragraph focuses on a comparison of CDI/colicin nuclease transport mechanisms.

2. Lines 35-36. I had difficulties to understand this sentence, specifically this phrase: “...that contain one N-terminal 36 residue too many, or too few,...”. Should it be a comma before “too many”? Even with that, I am not sure, I understand the meaning.

We have removed this phrase from the abstract, but added a similar sentence: “CdiA-CT^{EC3006} fragments fail to enter the cytoplasm if they contain even one residue too many (Δ VEN), or too few (Δ VENNY), at the N-terminus” at lines 108-109 of the revision. Hopefully, the revised sentence is more straightforward.

3. Please explain why polymyxin B E. coli cell permeabilization is needed.

CdiA-CT fragments are too large to cross the outer membrane and therefore cannot gain access to receptor in the cytoplasmic membrane (see Fig. 1f). This is consistent with our current understanding of CDI toxin delivery as illustrated in Fig. 1b. The CdiA-CT region is only transferred into the target-cell periplasm while still covalently linked to the full-length CdiA protein. We have revised the Results (lines 147-150) to clarify why polymyxin is required in these experiments.

4. Figs. 3A and 6A. What is the reason you chose presenting SEM instead of SDs?

We present standard errors in two instances (Figs. 3a & 7a) where we compare estimates of the population mean across treatment groups. All other values in the manuscript are standard deviations, which serve to indicate the spread of data points.

Reviewer #3 (Remarks to the Author):

The manuscript by Bartelli et al. elucidates an important aspect of CdiA toxin delivery during the antagonistic process of contact-dependent growth inhibition (CDI) by bacteria. During CDI, the protein CdiA delivers its C-terminal toxin domain across the outer and inner membranes of a recipient bacterium. This process is known to occur in a specific stepwise manner that requires (1) specific protein domains within CdiA, (2) specific membrane proteins on the recipient cell, and (3) proteolytic processing of the C-terminal region (CT) prior to inner membrane translocation. In this manuscript, the authors elegantly describe how this third point, proper proteolytic processing of the CT, shifts a previously stable cytoplasmic entry domain into a dynamic molten globule structure. This specific processing is required for delivery of the toxin across the inner membrane. Furthermore, their data supports earlier observations that this translocation step requires specific inner membrane protein receptors in the recipient cell and an intact proton gradient. The authors also present strong data suggesting that proteolytic processing of the CT occurs in the periplasm of the recipient bacterium and that this processing step is independent of the inner membrane receptors or proton gradient.

Throughout the manuscript, the authors use a robust series of experiments to analyze how CT processing, folding and translocation impact toxin delivery and ultimate growth inhibition. They are very thorough in their controls and their experiments were thoughtfully designed and elegantly performed. Although I do not have much experience in NMR spectroscopy, their explanation of the approaches and interpretations of the results were very clear. Overall, I found the manuscript to be extremely well-written and enjoyed reviewing the document. It is my opinion that this work is highly significant to the field of CDI biology. There is very little understanding of the CdiA transport processes, and this work provides valuable mechanistic insight into how this occurs. I did not observe any major flaws in their data analysis, interpretation, or conclusions. I applaud the authors for such fine work.

We thank the Reviewer for the kind comments, and hope that they remain supportive of the revised manuscript.

REVIEWERS' COMMENTS

Reviewer #1 (Remarks to the Author):

The authors have pursued my suspicion that the putative state responsible for initiation of transport was not really a classic molten globule. They has confirmed that is the case. Nevertheless, despite lacking the sexy MG moniker the plasticity of the state that triggers the transport remains highly interesting. As before, the paper is rigorous, complete and thoughtful in the presentation of an important biological process. The authors have nicely addressed the comments of the other referees. I therefore strongly encourage publication without further revision.

Reviewer #3 (Remarks to the Author):

The manuscript by Bartelli et al. elucidates an important aspect of CdiA toxin delivery during the antagonistic process of contact-dependent growth inhibition (CDI) by bacteria. During CDI, the protein CdiA delivers its C-terminal toxin domain across the outer and inner membranes of a recipient bacterium. This process is known to occur in a specific stepwise manner that requires (1) specific protein domains within CdiA, (2) specific membrane proteins on the recipient cell, and (3) proteolytic processing of the C-terminal region (CT) before inner membrane translocation. In this manuscript, the authors elegantly describe how this third point, proper proteolytic processing of the CT, shifts a previously stable cytoplasmic entry domain into a dynamic molten globule structure. This specific processing is required to deliver the toxin across the inner membrane. Furthermore, their data support earlier observations that this translocation step requires specific inner membrane protein receptors in the recipient cell and an intact proton gradient. The authors also present strong data suggesting that proteolytic processing of the CT occurs in the periplasm of the recipient bacterium and that this processing step is independent of the inner membrane receptors or proton gradient.

After considering comments from the other reviewers, assessing the authors rebuttal statements, and reviewing the revised manuscript, I stand with my initial review and have no major concerns with the manuscript. I only have minor edits to the document.

1. Line 78. The sentence "During biogenesis, CdiA is exported..." The phrase "is exported" should be changed to simply "export" to make it a correct sentence.

Reviewer #1 (Remarks to the Author):

The authors have pursued my suspicion that the putative state responsible for initiation of transport was not really a classic molten globule. They has confirmed that is the case. Nevertheless, despite lacking the sexy MG moniker the plasticity of the state that triggers the transport remains highly interesting. As before, the paper is rigorous, complete and thoughtful in the presentation of an important biological process. The authors have nicely addressed the comments of the other referees. I therefore strongly encourage publication without further revision.

We thank Reviewer #1 for their endorsement of our study.

Reviewer #3 (Remarks to the Author):

The manuscript by Bartelli et al. elucidates an important aspect of CdiA toxin delivery during the antagonistic process of contact-dependent growth inhibition (CDI) by bacteria. During CDI, the protein CdiA delivers its C-terminal toxin domain across the outer and inner membranes of a recipient bacterium. This process is known to occur in a specific stepwise manner that requires (1) specific protein domains within CdiA, (2) specific membrane proteins on the recipient cell, and (3) proteolytic processing of the C-terminal region (CT) before inner membrane translocation. In this manuscript, the authors elegantly describe how this third point, proper proteolytic processing of the CT, shifts a previously stable cytoplasmic entry domain into a dynamic molten globule structure. This specific processing is required to deliver the toxin across the inner membrane. Furthermore, their data support earlier observations that this translocation step requires specific inner membrane protein receptors in the recipient cell and an intact proton gradient. The authors also present strong data suggesting that proteolytic processing of the CT occurs in the periplasm of the recipient bacterium and that this processing step is independent of the inner membrane receptors or proton gradient.

After considering comments from the other reviewers, assessing the authors rebuttal statements, and reviewing the revised manuscript, I stand with my initial review and have no major concerns with the manuscript. I only have minor edits to the document.

1. Line 78. The sentence "During biogenesis, CdiA is exported..." The phrase "is exported" should be changed to simply "export" to make it a correct sentence.

We have corrected this grammatical error.